# ON DYNAMIC NOISE INFLUENCE IN DIFFERENTIAL PRIVATE LEARNING

## ABSTRACT

Protecting privacy in learning while maintaining the model performance has become increasingly critical in many applications that involve sensitive data. Private Gradient Descent (PGD) is a commonly used private learning framework, which noises gradients based on the Differential Privacy protocol. Recent studies show that *dynamic privacy schedules* of decreasing noise magnitudes can improve loss at the final iteration, and yet theoretical understandings of the effectiveness of such schedules and their connections to optimization algorithms remain limited. In this paper, we provide comprehensive analysis of noise influence in dynamic privacy schedules to answer these critical questions. We first present a dynamic noise schedule minimizing the utility upper bound of PGD, and show how the noise influence from each optimization step collectively impacts utility of the final model. Our study also reveals how impacts from dynamic noise influence change when momentum is used. We empirically show the connection exists for general non-convex losses, and the influence is greatly impacted by the loss curvature.

## 1 INTRODUCTION

In the era of big data, privacy protection in machine learning systems is becoming a crucial topic as increasing personal data involved in training models (Dwork et al., 2020) and the presence of malicious attackers (Shokri et al., 2017; Fredrikson et al., 2015). In response to the growing demand, differential-private (DP) machine learning (Dwork et al., 2006) provides a computational framework for privacy protection and has been widely studied in various settings, including both convex and non-convex optimization (Wang et al., 2017; 2019; Jain et al., 2019).

One widely used procedure for privacy-preserving learning is the (Differentially) Private Gradient Descent (PGD) (Bassily et al., 2014; Abadi et al., 2016). A typical gradient descent procedure updates its model by the gradients of losses evaluated on the training data. When the data is sensitive, the gradients should be *privatized* to prevent excess privacy leakage. The PGD privatizes a gradient by adding controlled noise. As such, the models from PGD is expected to have a lower utility as compared to those from unprotected algorithms. In the cases where strict privacy control is exercised, or equivalently, a tight *privacy budget*, accumulating effects from highly-noised gradients may lead to unacceptable model performance. It is thus critical to design effective privatization procedures for PGD to maintain a great balance between utility and privacy.

Recent years witnessed a promising privatization direction that studies how to dynamically adjust the privacy-protecting noise during the learning process, i.e., *dynamic privacy schedules*, to boost utility under a specific privacy budget. One example is (Lee & Kifer, 2018), which reduced the noise magnitude when the loss does not decrease, due to the observation that the gradients become very small when approaching convergence, and a static noise scale will overwhelm these gradients. Another example is (Yu et al., 2019), which periodically decreased the magnitude following a predefined strategy, e.g., exponential decaying or step decaying. Both approaches confirmed the empirically advantages of decreasing noise magnitudes. Intuitively, the dynamic mechanism may coordinate with certain properties of the learning task, e.g., training data and loss surface. Yet there is no theoretical analysis available and two important questions remain unanswered: 1) *What is the form of utility-preferred noise schedules?* 2) *When and to what extent such schedules improve utility?*

To answer these questions, in this paper we develop a principled approach to construct dynamic schedules and quantify their utility bounds in different learning algorithms. Our contributions

Table 1: Comparison of utility upper bound using different privacy schedules. The algorithms are $T$-iteration $\frac{1}{2}R$-zCDP under the PL condition (unless marked with *). The $O$ notation in this table drops other ln terms. Unless otherwise specified, all algorithms terminate at step $T = \mathcal{O}(\ln \frac{N^2 R}{D})$. Assume loss functions are 1-smooth and 1-Lipschitz continuous, and all parameters satisfy their numeric assumptions. Key notations: $\mathcal{O}_p$ – bound occurs in probability $p$; $D$ – feature dimension; $N$ – sample size; $R$ – privacy budget; $c_i$ – constant; other notations can be found in Section 4. An extended table and explanation are available in Appendix A.

| Algorithm | Schedule ($\sigma_t^2$) | Utility Upper Bound |
|---|---|---|
| GD+MA (Wang et al., 2017) | $\mathcal{O}(\frac{T}{R_{\epsilon,\delta}})$ | $\mathcal{O}\left(\frac{D \ln^2 N}{N^2 R_{\epsilon,\delta}}\right)$ |
| Adam+MA (Zhou et al., 2020) | $\mathcal{O}(\frac{T}{R_{\epsilon,\delta}})$ | $\mathcal{O}_p\left(\frac{\sqrt{D}\ln(ND\epsilon/(1-p))}{N R_{\epsilon,\delta}}\right)$ |
| GD, Non-Private | $0$ | $\mathcal{O}\left(\frac{D}{N^2 R}\right)$ |
| GD+zCDP, Static Schedule | $\frac{T}{R}$ | $\mathcal{O}\left(\frac{D \ln N}{N^2 R}\right)$ |
| GD+zCDP, Dynamic Schedule | $\mathcal{O}\left(\frac{\gamma^{(t-T)/2}}{R}\right)$ | $\mathcal{O}\left(\frac{D}{N^2 R}\right)$ |
| Momentum+zCDP, Static Schedule | $\frac{T}{R}$ | $\mathcal{O}\left(\frac{D}{N^2 R}(c + \ln N \mathbb{I}_{T>\hat{T}})\right)$ |
| Momentum+zCDP, Dynamic Schedule | $\mathcal{O}\left(\frac{c_1\gamma^{T+t}+c_2\gamma^{(T-t)/2}}{R}\right)$ | $\mathcal{O}\left(\frac{D}{N^2 R}(1 + \frac{cD}{N^2 R}\mathbb{I}_{T>\hat{T}})\right)$ |

are summarized as follows. 1) For the class of loss functions satisfying the Polyak-Lojasiewicz condition (Polyak, 1963), we show that a dynamic schedule improving the utility upper bound is shaped by the influence of per-iteration noise on the final loss. As the influence is tightly connected to the loss curvature, the advantage of using dynamic schedule depends on the loss function consequently. 2) Beyond gradient descent, our results show the gradient methods with momentum implicitly introduce a dynamic schedule and result in an improved utility bound. 3) We empirically validate our results on convex and non-convex (no need to satisfy the PL condition) loss functions. Our results suggest that the preferred dynamic schedule admits the exponentially decaying form, and works better when learning with high-curvature loss functions. Moreover, dynamic schedules give more utility under stricter privacy conditions (e.g., smaller sample size and less privacy budget).

## 2 RELATED WORK

**Differentially Private Learning**. Differential privacy (DP) characterizes the chance of an algorithm output (e.g., a learned model) to leak private information in its training data when the output distribution is known. Since outputs of many learning algorithms have undetermined distributions, the probability of their privacy leakages is hard to measure. A common approach to tackle this issue is to inject randomness with known probability distribution to *privatize* the learning procedures. Classical methods include output perturbation (Chaudhuri et al., 2011), objective perturbation (Chaudhuri et al., 2011) and gradient perturbation (Abadi et al., 2016; Bassily et al., 2014; Wu et al., 2017). Among these approaches, the Private Gradient Descent (PGD) has attracted extensive attention in recent years because it can be flexibly integrated with variants of gradient-based iteration methods, e.g., stochastic gradient descent, momentum methods (Qian, 1999), and Adam (Kingma & Ba, 2014), for both convex and non-convex problems.

**Dynamic Policies for Privacy Protection**. Wang et al. (2017) studied the empirical risk minimization using dynamic variation reduction of perturbed gradients. They showed that the utility upper bound can be achieved by gradient methods under uniform noise parameters. Instead of enhancing the gradients, Yu et al. (2019); Lee & Kifer (2018) showed the benefits of using a dynamic schedule of privacy parameters or equivalently noise scales. In addition, adaptive sensitivity control (Pichapati et al., 2019; Thakkar et al., 2019) and dynamic batch sizes (Feldman et al., 2020) are also demonstrated to improves the convergence.

**Utility Upper Bounds**. A utility upper bound is a critical metric for privacy schedules that characterizes the maximum utility that a schedule can deliver in theory. Wang et al. (2017) is the first to prove the utility bound under the PL condition. In this paper, we improve the upper bound by a more accurate estimation of the dynamic influence of step noise. Remarkably, by introducing a dynamic schedule, we further boost the sample-efficiency of the upper bound. With a similar intuition, Feldman et al. (2020) proposed to gradually increase the batch size, which reduces the dependence

on sample size accordingly. Recently, Zhou et al. proved the utility bound by using the momentum of gradients (Polyak, 1964; Kingma & Ba, 2014). Table 1 summarizes the upper bounds of methods studied in this paper (in the last block of rows) and results from state-of-the-art algorithms based on private gradients. Our work shows that considering the dynamic influence can lead to a tighter bound.

## 3 PRIVATE GRADIENT DESCENT

**Notations**. We consider a learning task by empirical risk minimization (ERM) $f(\theta) = \frac{1}{N}\sum_{n=1}^{N} f(\theta; x_n)$ on a private dataset $\{x_n\}_{n=1}^{N}$ and $\theta \in \mathbb{R}^D$. The gradient methods are defined as $\theta_{t+1} = \theta_t - \eta_t \nabla_t$, where $\nabla_t = \nabla f(\theta_t) = \frac{1}{N}\sum_n \nabla f(\theta_t; x_n)$ denotes the non-private gradient at iteration $t$, $\eta_t$ is the step learning rate. $\nabla_t^{(n)} = \nabla f(\theta_t; x_n)$ denotes the gradient on a sample $x_n$. $\mathbb{I}_c$ denotes the indicator function that returns $1$ if the condition $c$ holds, otherwise $0$.

**Assumptions**. (1) In this paper, we assume $f(\theta)$ is continuous and differentiable. Many commonly used loss functions satisfy this assumption, e.g., the logistic function. (2) For a learning task, only finite amount of privacy cost is allowed where the maximum cost is called *privacy budget* and denoted as $R$. (3) Generally, we assume that loss functions $f(\theta; x)$ (sample-wise loss) are $G$-Lipschitz continuous and $f(\theta)$ (the empirical loss) is $M$-smooth.

**Definition 3.1** (*$G$-Lipschitz continuity*). A function $f(\cdot)$ is $G$-Lipschitz continuous if, for $G > 0$ and all $x, y$ in the domain of $f(\cdot)$, $f(\cdot)$ satisfies $\|f(y) - f(x)\| \le G\|y - x\|^2$. .

**Definition 3.2** (*$m$-strongly convexity*). A function $f(\cdot)$ is $m$-strongly convex if $f(y) \ge f(x) + \nabla f(x)^T(y-x) + \frac{m}{2}\|y-x\|^2$, for some $m > 0$ and all $x, y$ in the domain of $f(\cdot)$.

**Definition 3.3** (*$M$-smoothness*). A function is $M$-smooth w.r.t. $l_2$ norm if $f(y) \le f(x) + \nabla f(x)^T(y-x) + \frac{M}{2}\|y-x\|^2$, for some constant $M > 0$ and all $x, y$ in the domain of $f(\cdot)$.

For a private algorithm $\mathcal{M}(d)$ which maps a dataset $d$ to some output, the privacy cost is measured by the bound of the output difference on the adjacent datasets. *Adjacent datasets* are defined to be datasets that only differ in one sample. In this paper, we use the zero-Concentrated Differential Privacy (zCDP, see Definition 3.4) as the privacy measurement, because it provides the simplicity and possibility of adaptively composing privacy costs at each iteration. Various privacy metrics are discussed or reviewed in (Desfontaines & Pejó, 2019). A notable example is Moment Accoutant (MA) (Abadi et al., 2016), which adopts similar principle for composing privacy costs while is less tight for a smaller privacy budget. We note that alternative metrics can be adapted to our study without major impacts to the analysis.

**Definition 3.4** (*$\rho$-zCDP* (Bun & Steinke, 2016)). Let $\rho > 0$. A randomized algorithm $\mathcal{M} : \mathcal{D}^n \to \mathbb{R}$ satisfies $\rho$-zCDP if, for all adjacent datasets $d, d' \in \mathcal{D}^n$, $D_\alpha(\mathcal{M}(d)\|\mathcal{M}(d')) \le \rho\alpha$, $\forall \alpha \in (1, \infty)$ where $D_\alpha(\cdot\|\cdot)$ denotes the Rényi divergence (Rényi, 1961) of order $\alpha$.

zCDP provides a linear composition of privacy costs of sub-route algorithms. When the input vector is privatized by injecting Gaussian noise of $\mathcal{N}(0, \sigma_t^2 I)$ for the $t$-th iteration, the composed privacy cost is proportional to $\sum_t \rho_t$ where the step cost is $\rho_t = \frac{1}{\sigma_t^2}$. For simplicity, we absorb the constant coefficient into the (residual) *privacy budget* $R$. The formal theorems for the privacy cost computation of composition and Gaussian noising is included in Lemmas B.1 and B.2.

Generally, we define the Private Gradient Descent (PGD) method as iterations for $t = 1 \ldots T$:

$$\theta_{t+1} = \theta_t - \eta_t \phi_t = \theta_t - \eta_t(\nabla_t + \sigma_t G\nu_t/N), \tag{1}$$

where $\phi_t = g_t$ is the gradient privatized from $\nabla_t$ as shown in Algorithm 1, $G/N$ is the bound of sensitivity of the gathered gradient excluding one sample gradient, and $\nu_t \sim \mathcal{N}(0, I)$ is a vector element-wisely subject to Gaussian distribution. We use $\sigma_t$ to denote the noise scale at step $t$ and use $\sigma$ to collectively represents the schedule $(\sigma_1, \ldots, \sigma_T)$ if not confusing. When the Lipschitz constant is unknown, we can control the upper bound by scaling the gradient if it is over some constant. The scaling operation is often called *clipping* in literatures since it clips the gradient norm at a threshold. After the gradient is noised, we apply a modification, $\phi(\cdot)$, to enhance its utility. In this paper, we consider two types of $\phi(\cdot)$:

$$\phi(m_t, g_t) = g_t \text{ (GD)}, \quad \phi(m_t, g_t) = [\beta(1 - \beta^{t-1})m_t + (1 - \beta)g_t]/(1 - \beta^t) \text{ (Momentum)}$$

We now show that the PGD using Algorithm 1 guarantees a privacy cost less than $R$:

---

**Algorithm 1** Privatizing Gradients

---

**Input**: Raw gradients $[\nabla_t^{(1)}, \ldots, \nabla_t^{(n)}]$ ($n = N$ by default), $v_t$, residual privacy budget $R_t$ assuming the full budget is $R$ and $R_1 = R$.

1: $\rho_t \leftarrow 1/\sigma_t^2, \nabla_t \leftarrow \frac{1}{n} \sum_{i=1}^n \nabla_t^{(i)}$          ▷ Budget request
2: **if** $\rho_t < R_t$ **then**
3:      $R_{t+1} \leftarrow R_t - \rho_t$
4:      $g_t \leftarrow \nabla_t + G\sigma_t \nu_t/N, \nu_t \sim \mathcal{N}(0, I)$          ▷ Privacy noise
5:      $m_{t+1} \leftarrow \phi(m_t, g_t)$ or $g_1$ if $t = 1$
6:      **return** $\eta_t m_{t+1}, R_{t+1}$          ▷ Utility projection
7: **else**
8:      Terminate

---

**Theorem 3.1.** *Suppose $f(\theta; x)$ is $G$-Lipschitz continuous and the PGD algorithm with privatized gradients defined by Algorithm 1, stops at step $T$. The PGD algorithm outputs $\theta_T$ and satisfies $\rho$-zCDP where $\rho \leq \frac{1}{2}R$.*

Note that Theorem 3.1 allows $\sigma_t$ to be different throughout iterations. Next we present a principled approach for deriving dynamic schedules optimized for the final loss $f(\theta_T)$.

# 4   DYNAMIC POLICIES BY MINIMIZING UTILITY UPPER BOUNDS

To characterize the utility of the PGD, we adopt the Expected Excess Risk (EER), which notion is widely used for analyzing the convergence of random algorithms, e.g., (Bassily et al., 2014; Wang et al., 2017). Due to the presence of the noise and the limitation of learning iterations, optimization using private gradients is expected to reach a point with a higher loss (i.e., excess risk) as compared to the optimal solution without private protection. Define $\theta^* = \arg\min_\theta f(\theta)$, after Algorithm 1 is iterated for $T$ times in total, the EER gives the expected utility degradation:

$$\text{EER} = \mathbb{E}_\nu[f(\theta_{T+1})] - f(\theta^*).$$

Due to the variety of loss function and complexity of recursive iterations, an exact EER with noise is intractable for most functions. Instead, we study the worst case scenario, i.e., the upper bound of the EER, and our goal is to minimize the upper bound. For consistency, we call the upper bound of EER divided by the initial error as ERUB. Since the analytical form of EER is either intractable or complicated due to the recursive iterations of noise, studying the ERUB is a convenient and tractable alternative. The upper bound often has convenient functional forms which are (1) sufficiently simple, such that we can directly minimize it, and (2) closely related to the landscape of the objective depending on both the training dataset and the loss function. As a consequence, it is also used in previous PGD literature (Pichapati et al., 2019; Wang et al., 2017) for choosing proper parameters. Moreover, we let $\text{ERUB}_{\min}$ be the achievable optimal upper bound by a specific choice of parameters, e.g., the $\sigma$ and $T$.

In this paper, we consider the class of loss functions satisfying the Polyak-Lojasiewicz (PL) condition which bounds losses by corresponding gradient norms. It is more general than the $m$-strongly convexity. If $f$ is differentiable and $M$-smooth, then $m$-strongly convexity implies the PL condition.

**Definition 4.1** (Polyak-Lojasiewicz condition (Polyak, 1963))**.** For $f(\theta)$, there exists $\mu > 0$ and for every $\theta$, $\|\nabla f(\theta)\|^2 \geq 2\mu(f(\theta) - f(\theta^*))$.

The PL condition helps us to reveal how the influence of step noise propagates to the final excess error, i.e., EER. Though the assumption was also used previously in Wang et al. (2017); Zhou et al. (2020), neither did they discuss the propagated influence of noise. In the following sections, we will show how the influence can tighten the upper bound in gradient descent and its momentum variant.

### 4.1 GRADIENT DESCENT METHODS

For the brevity of our discussion, we first define the following constants:

$$\frac{1}{\alpha} \triangleq \frac{2RMN^2}{DG^2}(f(\theta_1) - f(\theta^*)), \ \kappa \triangleq \frac{M}{\mu}, \text{ and } \gamma \triangleq 1 - \frac{1}{\kappa}, \tag{2}$$

which satisfy $\kappa \geq 1$ and $\gamma \in [0,1)$. Note that $\kappa$ is the condition number of $f(\cdot)$ if $f(\cdot)$ is strongly convex. $\kappa$ tends to be large if the function is sensitive to small differences in inputs, and $1/\alpha$ tends to be large if more samples are provided and with a less strict privacy budget. The convergence of PGD under the PL condition has been studied for private (Wang et al., 2017) and non-private (Karimi et al., 2016; Nesterov & Polyak, 2006; Reddi et al., 2016) ERM. Below we extend the bound in (Wang et al., 2017) by considering dynamic influence of noise and relax $\sigma_t$ to be dynamic:

**Theorem 4.1.** *Let $\alpha$, $\kappa$ and $\gamma$ be defined in Eq. (2), and $\eta_t = \frac{1}{M}$. Suppose $f(\theta; x_i)$ is G-Lipschitz and $f(\theta)$ is M-smooth satisfying the Polyak-Lojasiewicz condition. For PGD, the following holds:*

$$\text{ERUB} = \gamma^T + R \sum_{t=1}^{T} q_t \sigma_t^2, \text{ where } q_t \triangleq \gamma^{T-t} \alpha. \tag{3}$$

In Eq. (3), the step noise magnitude $\sigma_t^2$ has an exponential *influence*, $q_t$, on the EER. The dynamic characteristic of the influence is the key to prove a tighter bound. Plus, on the presence of the dynamic influence, it is natural to choose a dynamic $\sigma_t^2$. When relaxing $q_t$ to a static 1, a static $\sigma_t^2$ was studied by Wang et al. They proved a bound which is nearly optimal except a $\ln^2 N$ factor. To get the optimal bound, in the following sections, we look for the $\sigma$ and $T$ that minimize the upper bound.

#### 4.1.1 UNIFORM SCHEDULE

The uniform setting of $\sigma_t$ has been previously studied in Wang et al. (2017). Here, we show that the bound can be further tightened by considering the dynamic influence of iterations and a proper $T$.

**Theorem 4.2.** *Suppose conditions in Theorem 4.1 are satisfied. When $\sigma_t^2 = T/R$, let $\alpha$, $\gamma$ and $\kappa$ be defined in Eq. (2) and let $T$ be:*

$$T = \left\lceil \mathcal{O}\left(\kappa \ln\left(1 + \frac{1}{\kappa\alpha}\right)\right) \right\rceil. \tag{4}$$

*Meanwhile, if $\kappa \geq \frac{1}{1-c} > 1$, $1/\alpha > 1/\alpha_0$ for some constant $c \in (0,1)$ and $\alpha_0 > 0$, the corresponding bound is:*

$$\text{ERUB}_{\min}^{uniform} = \Theta\left(\frac{\kappa^2}{\kappa + 1/\alpha} \ln\left(1 + \frac{1}{\kappa\alpha}\right)\right). \tag{5}$$

*Sketch of proof.* The key of proof is to find a proper $T$ to minimize

$$\text{ERUB} = E = \gamma^T + \sum_{t=1}^{T} \gamma^{T-t} \alpha R \sigma^2 = \gamma^T + \alpha T \frac{1-\gamma^T}{1-\gamma} = \gamma^T + \alpha\kappa(1-\gamma^T)T$$

where we use $\sigma_t = \sqrt{T/R}$. Vanishing its gradient is to solve $\gamma^T \ln\gamma + \alpha\kappa(1-\gamma^T) - \alpha\kappa T\gamma^T \ln\gamma = 0$, which however is intractable. In (Wang et al., 2017), $T$ is chosen to be $\mathcal{O}(\ln(1/\alpha))$ and ERUB is relaxed as $\gamma^T + \alpha\kappa T^2$. The approximation results in a less tight bound as $\mathcal{O}(\alpha(1 + \kappa \ln^2(1/\alpha)))$ which explodes as $\kappa \to \infty$.

We observe that for a super sharp loss function, i.e., a large $\kappa$, any minor perturbation may result in tremendously fluctuating loss values. In this case, not-stepping-forward will be a good choice. Thus, we choose $T = \frac{1}{\ln(1/\gamma)} \ln\left(1 + \frac{\ln(1/\gamma)}{\alpha}\right) \leq \mathcal{O}\left(\kappa \ln\left(1 + \frac{1}{\kappa\alpha}\right)\right)$ which converges to 0 as $\kappa \to +\infty$. The full proof is deferred to the appendix. $\square$

#### 4.1.2 DYNAMIC SCHEDULE

A dynamic schedule can improve the upper bound delivered by the uniform schedule. First, we observe that the excess risk in Eq. (3) is upper bounded by two terms: the first term characterizes the error due to the finite iterations of gradient descents; the second term, a weighted sum, comes from error propagated from noise at each iteration. Now we show for any $\{q_t | q_t > 0, t = 1, \ldots, T\}$ (not limited to the $q_t$ defined in Eq. (3)), there is a unique $\sigma_t$ minimizing the weighted sum:

**Lemma 4.1** (Dynamic schedule). *Suppose $\sigma_t$ satisfy $\sum_{t=1}^{T} \sigma^{-2} = R$. Given a positive sequence $\{q_t\}$, the following equation holds:*

$$\min_{\sigma} R \sum_{t=1}^{T} q_t \sigma_t^2 = \left( \sum_{t=1}^{T} \sqrt{q_t} \right)^2, \text{ when } \sigma_t^2 = \frac{1}{R} \sum_{i=1}^{T} \sqrt{\frac{q_i}{q_t}}. \tag{6}$$

*Remarkably, the difference between the minimum and $T \sum_{t=1}^{T} q_t$ (uniform $\sigma_t$) monotonically increases by the variance of $\sqrt{q_t}$ w.r.t. $t$.*

We see that the dynamics in $\sigma_t$ come from the non-uniform nature of the weight $q_t$. Since $q_t$ presents the impact of the $\sigma_t$ on the final error, we denote it as *influence*. Given the dynamic schedule in Eq. (6), it is of our interest to which extent the ERUB can be improved. First, we present Theorem 4.3 to show the optimal $T$ and ERUB.

**Theorem 4.3.** *Suppose conditions in Theorem 4.1 are satisfied. Let $\alpha$, $\kappa$ and $\gamma$ be defined in Eq. (2). When $\eta_t = \frac{1}{M}$, $\sigma_t$ (based on Eqs. (3) and (6)) and the $T$ minimizing ERUB are, i.e.,*

$$\sigma_t^2 = \frac{1}{R} \frac{\sqrt{(1/\gamma)^T} - 1}{1 - \sqrt{\gamma}} \sqrt{\gamma^t}, \ T = \left\lceil \left( 2\kappa \ln \left( 1 + \frac{1}{\kappa\alpha} \right) \right) \right\rceil. \tag{7}$$

*Meanwhile, when $\kappa \geq 1$ and $1/\alpha \geq 1/\alpha_0$ for some positive constant $\alpha_0$, the minimal bound is:*

$$\text{ERUB}_{\min}^{dynamic} = \Theta \left( \frac{\kappa^2}{\kappa^2 + 1/\alpha} \right). \tag{8}$$

### 4.1.3 Discussion

In Theorems 4.2 and 4.3, we present the tightest bounds for functions satisfying the PL condition, to our best knowledge. We further analyze the advantages of our bounds from two aspects: sample efficiency and robustness to sharp losses.

**Sample efficiency**. Since dataset cannot be infinitely large, it is critical to know how accurate the model can be trained privately with a limited number of samples. Formally, it is of interest to study when $\kappa$ is fixed and $N$ is large enough such that $\alpha \gg 1$. Then we have the upper bound in Eq. (5) as

$$\text{ERUB}_{\min}^{\text{uniform}} \leq \mathcal{O} \left( \kappa^2 \alpha \ln \left( \frac{1}{\kappa\alpha} \right) \right) \leq \tilde{\mathcal{O}} \left( \frac{DG^2 \ln(N)}{MN^2 R} \right), \tag{9}$$

where we ignore $\kappa$ and other logarithmic constants with $\tilde{\mathcal{O}}$ as done in Wang et al. (2017). As a result, we get a bound very similar to (Wang et al., 2017), except that $R$ is replaced by $R_{MA} = \epsilon^2 / \ln(1/\delta)$ using Moment Accountant. In comparison, based on Lemma B.3, $R = 2\rho = 2\epsilon + 4\ln(1/\delta) + 4\sqrt{\ln(1/\delta)(\epsilon + \ln(1/\delta))}$ if $\theta_T$ satisfies $\rho$-zCDP. Because $\ln(1/\delta) > 1$, it is easy to see $R = R_{zCDP} > R_{MA}$ when $\epsilon \leq 2\ln(1/\delta)$. As compared to the one reported in (Wang et al., 2017), our bound saved a factor of $\ln N$ and thus is require less sample to achieve the same accuracy. Remarkably, the saving is due to the maintaining of the influence terms as shown in the proof of Theorem 4.2.

Using the dynamic schedule, we have $\text{ERUB}_{\min}^{\text{dynamic}} \leq \mathcal{O}(\alpha) = \mathcal{O} \left( \frac{DG^2}{MN^2 R} \right)$, which saved another $\ln N$ factor in comparison to the one using the uniform schedule Eq. (9). As shown in Table 1, such advantage maintains when comparing with other baselines.

**Robustness**. Besides sample efficiency, we are also interested in robustness of the convergence under the presence of privacy noise. Because of the privacy noise, the convergence of private gradient descent will be unable to reach an ideal spot. Specifically, when the samples are noisy or have noisy labels, the loss curvature may be sharp. The sharpness also implies lower smoothness, i.e., a small $M$ or has a very small PL parameter. Thus, gradients may change tremendously at some steps especially in the presence of privacy noise. As illustrated in the left figure, the highly-curved loss function (the green curve) results in mean higher final loss (the red dashed line) than the flatten curve (purple and blue lines). Such changes have more critical impact when only a less number of iterations can be executed due to the privacy constraint. Assume $\alpha$ is some constant while $\kappa \gg 1/\alpha$, we immediately get:

$$\text{ERUB}_{\min}^{\text{uniform}} = \Theta \left( \kappa \ln \left( 1 + \frac{1}{\kappa\alpha} \right) \right) = \Theta \left( \frac{1}{\alpha} \right) \leq \mathcal{O} \left( \frac{MN^2 R}{DG^2} \right), \ \text{ERUB}_{\min}^{\text{dynamic}} = \Theta(1).$$

Both are robust, but the dynamic schedule has a smaller factor since $1/\alpha$ could be a large number. In addition, the factor implies that when more samples are used, the dynamic schedule is robuster.

## 4.2 GRADIENT DESCENT METHODS WITH MOMENTUM

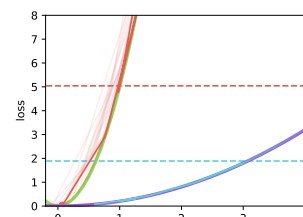

Section 4.1 shows that the step noise has an exponentially increasing influence on the final loss, and therefore a decreasing noise magnitude improves the utility upper bound by a $\ln N$ factor. However, the proper schedule can be hard to find when the curvature information, e.g., $\kappa$, is absent. A parameterized method that less depends on the curvature information is preferred. On the other hand, long-term iterations will result in forgetting of the initial iterations, since accumulated noise overwhelmed the propagated information from the beginning. This effect will reduce the efficiency of the recursive learning frameworks.

Figure 1: Private gradient descent repeated 100 times on two differently-curved loss functions. Solid lines are optimization trajectories and dashed horizontal lines are the averaged final losses.

Alternative to GD, the momentum method can mitigate the two issues. It was originally proposed to stabilize the gradient estimation (Polyak, 1964). In this section, we show that momentum (agnostic about the curvature) can flatten the dynamic influence and improve the utility upper bound. Previously, Pichapati et al. used the momentum as an estimation of gradient mean, without discussions of convergence improvements. Zhou et al. gave a bound for the Adam with DP. However, the derivation is based on gradient norm, which results in a looser bound (see Table 1).

The momentum method stabilizes gradients by moving average history coordinate values and thus greatly reduces the variance. The $\phi(m_t, g_t)$ can be rewritten as:

$$m_{t+1} = \phi(m_t, g_t) = \frac{v_{t+1}}{1 - \beta^t}, \ v_{t+1} = \beta v_t + (1 - \beta)g_t = (1 - \beta)\sum\nolimits_{i=1}^{t} \beta^{t-i}g_t, \ v_1 = 0, \quad (10)$$

where $\beta \in [0, 1]$. Note $v_{t+1}$ is a biased estimation of the gradient expectation while $m_{t+1}$ is unbiased.

**Theorem 4.4** (Convergence under PL condition). *Suppose $f(\theta; x_i)$ is $G$-Lipschitz, and $f(\theta)$ is $M$-smooth and satisfies the Polyak-Lojasiewicz condition. Assume $\beta \neq \gamma$ and $\beta \in (0, 1)$. Let $\eta_t = \frac{\eta_0}{2M}$ and $\eta_0 \leq 8\left(\sqrt{1 + 64\beta\gamma(\gamma - \beta)^{-2}(1 - \beta)^{-3}} + 1\right)^{-1}$. Then the following holds:*

$$\text{EER} \leq \left(\gamma^T + 2R\eta_0\alpha \underbrace{U_3(\sigma, T)}_{\text{noise varinace}}\right)(f(\theta_1) - f(\theta^*)) - \zeta\frac{\eta_0}{2M}\underbrace{\sum\nolimits_{t=1}^{T}\gamma^{T-t}\mathbb{E}\left\|v_{t+1}\right\|^2}_{\text{momentum effect}} \quad (11)$$

$$\text{where} \quad \gamma = 1 - \frac{\eta_0}{\kappa}, \ \zeta = 1 - \frac{1}{\beta(1 - \beta)^3}\eta_0^2 - \frac{1}{4}\eta_0 \geq 0, \quad (12)$$

$$U_3 = \sum\nolimits_{t=1}^{T}\gamma^{T-t}\frac{(1 - \beta)^2}{(1 - \beta^t)^2}\sum\nolimits_{i=1}^{t}\beta^{2(t-i)}\sigma_i^2. \quad (13)$$

The upper bound includes three parts that influence the bound differently: *(1) Convergence.* The convergence term is mainly determined by $\eta_0$ and $\kappa$. $\eta_0$ should be in $(0, \kappa)$ such that the upper bound can converge. A large $\eta_0$ will be preferred to speed up convergence if it does not make the rest two terms worse. *(2) Noise Variance.* The second term compressed in $U_3$ is the effect of the averaged noise, $\sum_{i=1}^{t}\beta^{2(t-i)}\sigma_i^2$. One difference introduced by the momentum is the factor $(1 - \beta)/(1 - \beta^t)$ which is less than $\gamma^t$ at the beginning and converges to a non-zero constant $1 - \beta$. Therefore, in $U_3$, $\gamma^{T-t}(1 - \beta)/(1 - \beta^t)$ will be constantly less than $\gamma^T$ meanwhile. Furthermore, when $t > \hat{T}$, the moving average $\sum_{i=1}^{t}\beta^{2(t-i)}\sigma_i^2$ smooths the influence of each $\sigma_t$. In Appendix D, we will see that the influence dynamics is less steep than that of GD. *(3) Momentum Effect.* The momentum effect term can improve the upper bound when $\eta_0$ is small. For example, when $\beta = 0.9$ and $\gamma = 0.99$, then $\eta_0 \leq 0.98/M$ which is a rational value. Following the analysis, when $M$ is large which means the gradient norms will significantly fluctuate, the momentum term may take the lead. Adjusting the noise scale in this case may be less useful for improving utility.

To give an insight on the effect of dynamic schedule, we provide the following utility bounds.

**Theorem 4.5** (Uniform schedule). *Suppose the assumptions in Theorem 4.4 are satisfied. Let $\sigma_t^2 = T/R$, and let:*

$$\hat{T} = \max t \ s.t. \ \gamma^{t-1} \geq \frac{1-\beta}{1-\beta^t}, \ T = \left\lceil \mathcal{O}\left(\frac{\kappa}{\eta_0}\ln\left(1 + \frac{\eta_0}{\kappa\alpha}\right)\right)\right\rceil.$$

*Given some positive constant $c$ and $\alpha_0 > 0$ with $1/\alpha > 1/\alpha_0$, the following inequality holds:*

$$\text{ERUB}_{\min} \leq \mathcal{O}\left(\frac{\kappa^2}{\kappa + \eta_0/\alpha}\left[\mathbb{I}_{T\leq\hat{T}} + \gamma^{\hat{T}-1}\ln\left(1 + \frac{\eta_0}{\kappa\alpha}\right)\mathbb{I}_{T>\hat{T}}\right]\right).$$

**Theorem 4.6** (Dynamic schedule). *Suppose the assumptions in Theorem 4.4 are satisfied. Let $\alpha' = \frac{2\eta_0\alpha}{\gamma(1-\gamma\beta^2)}$, $\beta < \gamma$ and $\hat{T} = \max t \ s.t. \ \gamma^{t-1} \geq \frac{1-\beta}{1-\beta^t}$. Use the following schedule:*

$$\sigma_t^2 = \frac{1}{R}\sum_{i=1}^{T}\sqrt{\frac{q_i}{q_t}}, \ T^{dyn} = \left\lceil\mathcal{O}\left(\frac{2\kappa}{\eta_0}\ln\left(1 + \frac{\eta_0}{\kappa\alpha}\right)\right)\right\rceil,$$

*where $q_t = c_1\gamma^{T+t}\mathbb{I}_{T\leq\hat{T}} + \gamma^{\hat{T}-1}c_2\gamma^{T-t}\mathbb{I}_{T>\hat{T}}$ for some positive constants $c_1$ and $c_2$. The following inequality holds:*

$$\text{ERUB} \leq \gamma^T + 2\eta_0\alpha\sum_{t=1}^{T}Rq_t\sigma_t^2, \ \text{ERUB}_{\min} \leq \mathcal{O}\left(\frac{\kappa\alpha}{\kappa\alpha + \eta_0}\left(\frac{\kappa\alpha}{\kappa\alpha + \eta_0}\mathbb{I}_{T\leq\hat{T}} + \mathbb{I}_{T>\hat{T}}\right)\right).$$

**Discussion.** Theoretically, the dynamic schedule is more influential in vanilla gradient descent methods than the momentum variant. The result is mainly attributed to the averaging operation. The moving averaging, $(1-\beta)\sum_{i=1}^{t}\beta^{t-i}g_i/(1-\beta^t)$, increase the influence of the under-presented initial steps and decrease the one of the over-sensitive last steps. Counterintuitively, the preferred dynamic schedule should be increasing since $q_t$ decreases when $t \leq \hat{T}$.

## 4.3 PRIVATE STOCHASTIC GRADIENT DESCENT (NEW SECTION ON REBUTTAL)

Though PGD provides a guarantee both for utility and privacy, computing gradients of the whole dataset is impractical for large-scale problems. For this sake, studying the convergence of Private Stochastic Gradient Descent (PSGD) is meaningful. The Algorithm 1 can be easily extended to PSGD by subsampling $n$ gradients where the batch size $n \ll N$. According to (Yu et al., 2019), when privacy is measured by zCDP, there are two ways to account for the privacy cost of PSGD depending on the batch-sampling method: sub-sampling with or without replacement. In this paper, we focus on the random subsampling with replacement since it is widely used in deep learning in literature, e.g., (Abadi et al., 2016; Feldman et al., 2020). Accordingly, we replace $N$ in the definition of $\alpha$ by $n$ because the term is from the sensitivity of batch data (see Eq. (1)). For clarity, we assume that $T$ is the number of iterations rather than epochs and that $\tilde{\nabla}_t$ is mean stochastic gradient.

When a batch of data are randomly sampled, the privacy cost of one iteration is $cp^2/\sigma_t$ where $c$ is some constant, $p = n/N$ is the sample rate, and $1/\sigma_t^2$ is the full-batch privacy cost. Details of the sub-sampling theorems are referred to the Theorem 3 of (Yu et al., 2019) and their empirical setting. Threfore, we can replace the privacy constraint $\sum_t p^2/\sigma_t^2 = R$ by $\sum_t 1/\sigma_t^2 = R'$ where $R' = R/p^2 = \frac{N^2}{n^2}R$. Remarkably, we omit the constant $c$ because it will not affect the results regarding uniform or dynamic schedules. Notice $N^2R$ in the $\alpha$ is replaced by $n^2R' = N^2R$. Thus, the form of $\alpha$ is not changed which provides convenience for the following derivations.

Now we study the utility bound of PSGD. To quantify the randomness of batch sampling, we define a random vector $\xi_t$ with $\mathbb{E}[\xi_t] = 0$ and $\mathbb{E}\|\xi_t\|^2 \leq D$ such that $\tilde{\nabla}_t \leq \nabla_t + \sigma_g\xi_t/n$ for some positive constant $\sigma_g$. Because $\xi_t$ has similar property to the privacy noise $\nu_t$, we can easily extend the PGD bounds to PSGD bounds by following theories.

**Theorem 4.7** (Utility bounds of PSGD). *Let $\alpha$, $\kappa$ and $\gamma$ be defined in Eq. (2), and $\eta_t = \frac{1}{M}$. Suppose $f(\theta; x_i)$ is $G$-Lipschitz and $f(\theta)$ is $M$-smooth satisfying the Polyak-Lojasiewicz condition. For PSGD, when batch size satisfies $n = \max\{N\sqrt{R}, 1\}$, the following holds:*

$$\text{ERUB} = \gamma^T + \alpha_g\sigma_g^2 + R'\sum_{t=1}^{T}q_t\sigma_t^2, \ \text{where } q_t \triangleq \gamma^{T-t}\alpha, \ \sum_t 1/\sigma_t^2 = R'. \qquad (14)$$

*where $\alpha_g = \frac{D}{2\mu N^2 R(f(\theta_1) - f(\theta^*))}$.*

**Theorem 4.8** (PSGD with momentum). *Let* $\alpha_g = \frac{D}{2\mu N^2 R(f(\theta_1) - f(\theta^*))}$. *Suppose assumptions in Theorem 4.4 holds. When batch size satisfies* $n = \max\{N\sqrt{R}, 1\}$, *the* $U_3(\sigma, T)$ *has to be replaced by*

$$\tilde{U}_3 = U_3^g + U_3, \text{ with } \alpha R' U_3^g \leq \alpha_g \sigma_g^2 \tag{15}$$

*when PSGD is used.*

As shown above, the utility bound of PSGD differs from the PGD merely by $\alpha_g \sigma_g^2$. Note $\alpha_g = \mathcal{O}(\frac{D}{N^2 R})$ which fits the order of dynamic-schedule bounds. In addition, $\alpha$ and other variables are not changed. Hence, the conclusions w.r.t. the dynamic/uniform schedules maintain the same.

## 5 EXPERIMENTS

We empirically validate the properties of privacy schedules and their connections to learning algorithms. In this section, we briefly review the schedule behavior on quadratic losses under varying data sensitivity. Details of experimental setups and empirical results are available in Appendix D.

We first show the estimated influence of step noise $q_t$ (by retraining the private learning algorithms, ref. Appendix D) in Fig. 2 Left. We see the trends of influence are approximately in an exponential form of $t$. This obvervation motivates the use of exponential decay schedule in practice.

We then show the trends on the variance of influence (dashed lines with the right axis) and relative final losses (solid lines with the left axis) in the Middle Pane, where *uni* denotes the uniform schedule baseline, *exp* is an exponential schedule, *dyn* denotes the dynamic schedule minimizing the ERUB. The influences increases steeply when the data scale is large and therefore have a large variance. Meanwhile, dynamic schedules show improvements of the final loss when the variance is large. It reveals the connection between the influence and the dynamic advantage (refer to Lemma 4.1).

We lastly evaluate the impacts from momentum in the Right Pane, using a Deep Neural Network (DNN) with 2 layers and 100 hidden units. Because of time costs of training deep networks, we do not estimate the influence by retraining and then compute schedules. Instead, we grid-search for the schedule hyper-parameters to find the best one. We see that influence modeled by an exponential function (*expinfl*) has comparable performance of the influence modeled by linear combination of two reverse exponential functions (*momexpinfl*). The latter only shows advantage in the setting that data scale is 25 and the number of iteration is only 100, which is expected by our analysis Theorems 4.5 and 4.6. The inherent reason is that the dynamic schedule is more effective when $T$ is larger.

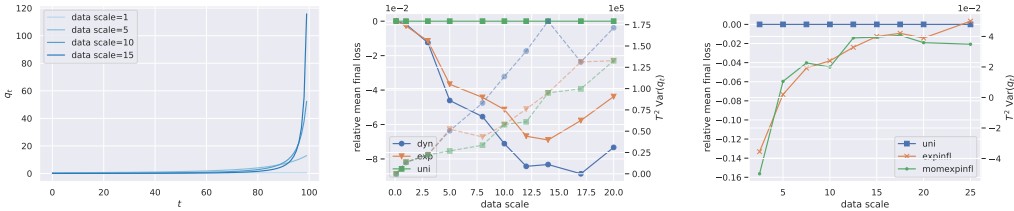

Figure 2: Comparison of dynamic schedule and uniform schedule on different data scale. Left pane is the influence by iteration estimated by retraining. The rest two are the relative loss by varying data scale (left axis with solid lines) and the variance of influence (right axis with dashed lines). The middle is on the MNIST35 dataset consisting of 1000 digit 3 and 5 images using quadratic regression. The right is the final loss on subsampled MNIST dataset of 1000 training samples and 50,000 test samples when using DNN and momentum methods.

## 6 CONCLUSION

When a privacy budget is provided for a certain learning task, one has to carefully schedule the privacy usage through the learning process. Uniformly scheduling the budget has been widely used in literature whereas increasing evidence suggests that dynamically schedules could empirically outperform the uniform one. This paper provided a principled analysis on the problem of optimal budget allocation and connected the advantages of dynamic schedules to both the loss structure and the learning behavior. We further validated our results through empirical studies.

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

## A  COMPARISON OF ALGORITHMS

Table 2: Comparison of empirical excess risk bounds. The algorithms are $T$-iteration $\frac{1}{2}R$-zCDP or equivalently $(\epsilon, \delta)$-DP under the PL condition (unless marked with * for convexity). The $\mathcal{O}$ notation in this table drops other $\ln$ terms. All algorithms in the second part terminate at step $T = \mathcal{O}(\ln \frac{N^2 R}{D})$. Assume loss functions are 1-smooth and 1-Lipschitz continuous, and all parameters satisfy their numeric assumptions. Key notations: $\mathcal{O}_p$ – bound occurs in probability $p$; $D$ – feature dimension; $N$ – sample size; $R$ – privacy budget; $c_i$ – constant.

| Algorithm | Schedule ($\sigma_t^2$) | Utility Upper Bd. |
|---|---|---|
| *GD+Adv (Bassily et al., 2014) | $\mathcal{O}\left(\frac{\ln(N/\delta)}{R_{\epsilon,\delta}}\right)$ | $\mathcal{O}\left(\frac{D \ln^3 N}{N R_{\epsilon,\delta}}\right)$ |
| GD+MA (Wang et al., 2017) | $\mathcal{O}(\frac{T}{R_{\epsilon,\delta}})$ | $\mathcal{O}\left(\frac{D \ln^2 N}{N^2 R_{\epsilon,\delta}}\right)$ |
| *GD+Adv+BBImp (Cummings et al., 2018) | $\mathcal{O}\left(\frac{n^2 \ln(n/\delta)}{R_{\epsilon,\delta}}\right)$ | $\mathcal{O}_p\left(\frac{D^2 \ln^2(1/p)}{R_{\epsilon,\delta} N^{1-c}}\right)$ |
| Adam+MA (Zhou et al., 2020) | $\mathcal{O}(\frac{T}{R_{\epsilon,\delta}})$ | $\mathcal{O}_p\left(\frac{\sqrt{D} \ln(ND\epsilon/(1-p))}{N R_{\epsilon,\delta}}\right)$ |
| GD, Non-Private | $0$ | $\mathcal{O}\left(\frac{D}{N^2 R}\right)$ |
| GD+zCDP, Static Schedule | $\frac{T}{R}$ | $\mathcal{O}\left(\frac{D \ln N}{N^2 R}\right)$ |
| GD+zCDP, Dynamic Schedule | $\mathcal{O}\left(\frac{\gamma^{(t-T)/2}}{R}\right)$ | $\mathcal{O}\left(\frac{D}{N^2 R}\right)$ |
| Momentum+zCDP, Static Schedule | $\frac{T}{R}$ | $\mathcal{O}\left(\frac{D}{N^2 R}(c + \ln N \mathbb{I}_{T>\hat{T}})\right)$ |
| Momentum+zCDP, Dynamic Schedule | $\mathcal{O}\left(\frac{c_1 \gamma^{T+t} + c_2 \gamma^{(T-t)/2}}{R}\right)$ | $\mathcal{O}\left(\frac{D}{N^2 R}(1 + \frac{cD}{N^2 R}\mathbb{I}_{T>\hat{T}})\right)$ |

We present Table 2 as an sumpplementary to the Table 1. Asymptotic upper bounds are achieved when sample size $N$ approaches infinity. Both $R$ and $R_{\epsilon,\delta}$ with $R_{\epsilon,\delta} < R$ are the privacy budgets of corresponding algorithms. Specifically, $R_{\epsilon,\delta} = \epsilon^2/\ln(1/\delta) < R$ when the private algorithm is $(\epsilon, \delta)$-DP with $\epsilon \leq 2\ln(1/\delta)$.

**PGD+Adv**. Adv denotes the Advanced Composition method (Bassily et al., 2014). The method assumes that loss function is 1-strongly convex which implies the PL condition and optimized variable is in a convex set of diameter 1 w.r.t. $l_2$ norm.

**PGD+MA**. MA denotes the Moment Accoutant (Abadi et al., 2016) which improve the composed privacy bound versus the Advanced Composition. The improvement on privacy bound lead to a enhanced utility bound, as a result.

**PGD+Adv+BBImp**. The dynamic method assumes that the loss is 1-strongly convex and data comes in stream with $n \leq N$ samples at each round. Their utility upper bound is achieved at some probability $p$ with any positive $c$.

**Adam+MA**. The authors prove a convergence bound for the gradient norms which is extended to loss bound by using PL condition. They also presents the results for AdaGrad and GD which are basically of the same upper bound. Out theorems improve their bound by using the recursive derivation based on the PL condition, while their bound is a simple application of the condition on the gradient norm bound.

**GD, Non-Private**. This method does not inject noise into gradients but limit the number of iterations. With the bound, we can see that our utility bound are optimal with dynamic schedule.

**GD+zCDP**. We discussed the static and dynamic schedule for the gradient descent method where the dynamic noise influence is the key to tighten the bound.

**Momemtum+zCDP**. Different from the GD+zCDP, momentum methods will have two phase of utility upper bound. When $T$ is small than some positive constant $\hat{T}$, the bound is as tight as the non-private one. Afterwards, the momentum has a bound degraded as the GD bound.

Table 3: Comparison of true excess risk bounds. The algorithms are $T$-iteration $\frac{1}{2}R$-zCDP or equivalently $(\epsilon, \delta)$-DP under the $\mu$-strongly-convex condition. The $\mathcal{O}$ notation in this table drops other $\ln$ terms. Assume loss functions are $1$-smooth and $1$-Lipschitz continuous, and all parameters satisfy their numeric assumptions. * marks the method with convex assumption.

| Algorithm | Utility Upper Bd. | T |
|---|---|---|
| GD+Adv (Bassily et al., 2014) | $\mathcal{O}_{1-p}\left(\frac{\sqrt{D}\ln^2 N \ln(1/p)}{p\mu N R_{\epsilon,\delta}}\right)$ | $N^2$ |
| SVRG+MA (Wang et al., 2017) | $\mathcal{O}\left(\frac{D\ln N}{\mu N^2 R_{\epsilon,\delta}}\right)$ | $\mathcal{O}(\ln \frac{N^2 R_{\epsilon,\delta}}{D})$ |
| SSGD+zCDP (Feldman et al., 2020) | $\mathcal{O}\left(\left(\frac{1}{\sqrt{N}} + \frac{2\sqrt{D}}{\sqrt{R}N}\right)\ln N\right)$ | $\frac{N^2}{16D/R^2+4N}$ |
| * SGD+MA (Bassily et al., 2019) | $\mathcal{O}\left(\max\left\{\frac{\sqrt{D}}{N\sqrt{R_{\epsilon,\delta}}}, \frac{1}{\sqrt{N}}\right\}\right)$ | $\min\{\frac{N}{8}, \frac{N^2 R_{\epsilon,\delta}}{32D}\}$ |
| GD+zCDP, Static Schedule | $\mathcal{O}_{1-p}\left(\frac{G^2}{\mu N}\left(\sqrt{\frac{D\ln(N)\ln(1/p)}{NR}} + \frac{4}{p}\right)\right)$ | $\mathcal{O}(\ln \frac{N^2 R}{D})$ |
| GD+zCDP, Dynamic Schedule | $\mathcal{O}_{1-p}\left(\frac{G^2}{\mu N}\left(\sqrt{\frac{D\ln(1/p)}{NR}} + \frac{4}{p}\right)\right)$ | $\mathcal{O}(\ln \frac{N^2 R}{D})$ |
| Momentum+zCDP, Static Sch. | $\mathcal{O}_{1-p}\left(\frac{G^2}{\mu N}\left(\sqrt{\frac{D\ln(1/p)}{NR}(c + \ln N\mathbb{I}_{T>\hat{T}})} + \frac{4}{p}\right)\right)$ | $\mathcal{O}(\ln \frac{N^2 R}{D})$ |
| Momentum+zCDP, Dynamic Sch. | $\mathcal{O}_{1-p}\left(\frac{G^2}{\mu N}\left(\sqrt{\frac{D\ln(1/p)}{NR}(1 + \frac{cD}{N^2 R}\mathbb{I}_{T>\hat{T}})} + \frac{4}{p}\right)\right)$ | $\mathcal{O}(\ln \frac{N^2 R}{D})$ |
| GD, Non-Private | $\mathcal{O}\left(\frac{D}{N^2 R}\right)$ | $\mathcal{O}(\ln \frac{N^2 R}{D})$ |
| GD+zCDP, Static Schedule | $\mathcal{O}\left(\frac{D\ln N}{N^2 R}\right)$ | $\mathcal{O}(\ln \frac{N^2 R}{D})$ |
| GD+zCDP, Dynamic Schedule | $\mathcal{O}\left(\frac{D}{N^2 R}\right)$ | $\mathcal{O}(\ln \frac{N^2 R}{D})$ |
| Momentum+zCDP, Static Sch. | $\mathcal{O}\left(\frac{D}{N^2 R}(c + \ln N\mathbb{I}_{T>\hat{T}})\right)$ | $\mathcal{O}(\ln \frac{N^2 R}{D})$ |
| Momentum+zCDP, Dynamic Sch. | $\mathcal{O}\left(\frac{D}{N^2 R}(1 + \frac{cD}{N^2 R}\mathbb{I}_{T>\hat{T}})\right)$ | $\mathcal{O}(\ln \frac{N^2 R}{D})$ |

## A.1 COMAPRISON OF GENERALIZATION BOUNDS

In addition to the empirical risk bounds in Table 2, in this section we study the *true risk bounds*, or generalization error bounds. True risk bounds characterize how well the learnt model can generalize to unseen samples subject to the inherent data distribution. By leveraging the generic learning-theory tools, we extend our results to the *True Excess Risk* (TER) for strongly convex functions as follows. For a model $\theta$, its TER is defined as follows:

$$\text{TER} \triangleq \mathbb{E}_{x \sim \mathcal{X}}[\mathbb{E}[f(\theta; x)]] - \min_{\hat{\theta}} \mathbb{E}_{x \sim \mathcal{X}}[f(\hat{\theta}; x)],$$

where the second expectation is over the randomness of generating $\theta$ (e.g., the noise and stochastic batches). Assume a dataset $d$ consist of $N$ samples drawn i.i.d. from the distribution $\mathcal{X}$. Two approaches could be used to extend the empirical bounds to the true excess risk: One is proposed by Shalev-Shwartz et al. (2009) where the true excess risk of PGD can be bounded in high probability. For example, Bassily et al. (2014) achieved a $\frac{\ln^2 N}{N}$ bound with $N^2$ iterations. Alternatively, instead of relying on the probabilistic bound, Bassily et al. (2019) used the uniform stability to give a tighter bound. Later, Feldman et al. (2020) improve the efficiency of gradient computation to achieve a similar bound. Both approaches introduce an additive term to the empirical bounds. In this section, we adopt both approaches to investigate the two types of resulting true risk bounds.

**(1) True Risk in High Probability**. First, we consider the high-probability true risk bound. Based on Section 5.4 from (Shalev-Shwartz et al., 2009) (restated in Theorem A.1), we can relate the EER to the TER.

**Theorem A.1.** *Let $f(\theta; x)$ be $G$-Lipschitz, and $f(\theta)$ be $\mu$-strong convex loss function given any $x \in \mathcal{X}$. With probability at least $1 - p$ over the randomness of sampling the data set d, the following inequality holds:*

$$\text{TER}(\theta) \leq \sqrt{\frac{2G^2}{\mu N}}\sqrt{f(\theta) - f(\theta^*)} + \frac{4G^2}{p\mu N}, \tag{16}$$

*where $\theta^* = \arg\min_\theta f(\theta)$.*

To apply the Eq. (16), we need to extend EER, the expectation bound, to a high-probability bound. Following (Bassily et al., 2014) (Section D), we repeat the PGD with privacy budget $R/k$ for $k$ times. Note, the output of all repetitions is still of $R$ budget. When $k = 1$, let the EER of the algorithm be denoted as $F(R)$. Then the EER of one execution of the $k$ repetitions is $F(R/k)$ where privacy is accounted by zCDP. When $k = \log_2(1/p)$ for $p \in [0,1]$, by Markov's inequality, there exists one repetition whose EER is $F(R/\log_2(1/p))$ with probability at least $1 - 1/2^k = 1 - p$. Combined with Eq. (16), we use the bounds of uniform schedule and dynamic schedules in Section 4.1.3 to obtain:

$$\text{TER}^{\text{uniform}} \leq \tilde{\mathcal{O}}\left(\frac{G^2}{\mu N}\left(\sqrt{\frac{D\ln(N)\ln(1/p)}{NR}} + \frac{4}{p}\right)\right), \tag{17}$$

$$\text{TER}^{\text{dynamic}} \leq \tilde{\mathcal{O}}\left(\frac{G^2}{\mu N}\left(\sqrt{\frac{D\ln(1/p)}{NR}} + \frac{4}{p}\right)\right), \tag{18}$$

where we again ignore the $\kappa$ and other constants. Similarly, we can extend the momentum methods.

**(2) True Risk by Unfirom Stability**. Following Bassily et al. (2019), we use the uniform stability (defined in Definition A.1) to extend the empirical bounds. We restate the related definition and theorems as follows.

**Definition A.1** (Uniform stability). Let $s > 0$. A randomized algorithm $\mathcal{M} : \mathcal{D}^N \to \Theta$ is $s$-uniformly stable w.r.t. the loss function $f$ if for any neighor datasets $d$ and $d'$, we have:

$$\sup_{x \in \mathcal{X}} \mathbb{E}[f(\mathcal{M}(d); x) - f(\mathcal{M}(d'); x)] \leq s,$$

where the expectation is over the internal randomness of $\mathcal{M}$.

**Theorem A.2** (See, e.g., (Shalev-Shwartz & Ben-David, 2014)). *Suppose $\mathcal{M} : \mathcal{D}^N \to \Theta$ is a $s$-uniformly stable algorithm w.r.t. the loss function $f$. Let $\mathcal{D}$ be any distribution from over data space and let $d \sim \mathcal{D}^N$. The following holds true.*

$$\mathbb{E}_{d \sim \mathcal{D}^N}[\mathbb{E}[f(\mathcal{M}(d); \mathcal{D}) - f(\mathcal{M}(d); d)]] \leq s,$$

*where the second expectation is over the internal randomness of $\mathcal{M}$. $f(\mathcal{M}(d); \mathcal{D})$ and $f(\mathcal{M}(d); d)$ represent the true loss and the empirical loss, respecitvely.*

**Theorem A.3** (Uniform stability of PGD from (Bassily et al., 2019)). *Suppose $\eta < 2/M$ for $M$ smooth, $G$-Lipschitz $f(\theta; x)$. Then PGD is $s$-uniformly stable with $s = G^2 T\eta/N$.*

Combining Theorems A.2 and A.3, we obtain the following:

$$\text{TER} \leq \text{EER} + G^2 \frac{\eta T}{N}.$$

Because EER in this paper compresses a $\gamma^T$ or similar exponential terms, unlike (Bassily et al., 2019), we cannot directly minimize the TER upper bound w.r.t. $T$ and $\eta$ in the presence of a polynomial form of $\gamma^T$ and $T$. Therefore, we still use $T = \mathcal{O}(\ln \frac{N^2 R}{D})$ and $\eta$ for minimizing EER. Note that

$$G^2 \frac{\eta T}{N} \leq \mathcal{O}(\frac{G^2}{MN}\ln\frac{N^2 R}{D}) \leq \mathcal{O}\left(\frac{G^2}{M}\right)$$

where we assume $N \gg D$ and use $\ln N \leq N$. Because the term $\mathcal{O}\left(G^2/M\right)$ is constant and independent from dimension, we follow (Bassily et al., 2019) to drop the term when comparing the bounds. After dropping the additive term, it is obvious to see that the advantage of dynamic schedules still maintains since $\text{TER} \leq \text{EER}$. A similar extension can be derived for (Wang et al., 2017).

We summarize the results and compare them to prior works in Table 3 where we include an additional method: Snowball Stochastic Gradient Descent (SSGD). SSGD dynamically schedule the batch size to achieve an optimal convergence rate in linear time.

**Discussion**. By using uniform stability, we successfully transfer the advantage of our dynamic schedules from empirical bounds to true risk bounds. The inherent reason is that our bounds only need $\ln N$ iterations to reach the preferred final loss. With uniform stability, the logarithmic $T$ reduce the gap caused by transferring. Compared to the (Feldman et al., 2020; Bassily et al., 2019), our method has remarkably improved efficiency in $T$ from $N$ or $N^2$ to $\ln(N)$. That implies fewer iterations are required for converging to the same generalization error.

## B Preliminaries

### B.1 Privacy

**Lemma B.1** (Composition & Post-processing). *Let two mechanisms be $M : \mathcal{D}^n \to \mathcal{Y}$ and $M' : \mathcal{D}^n \times \mathcal{Y} \to \mathcal{Z}$. Suppose $M$ satisfies $(\rho_1, a)$-zCDP and $M'(\cdot, y)$ satisfies $(\rho_2, a)$-zCDP for $\forall y \in \mathcal{Y}$. Then, mechanism $M'' : \mathcal{D}^n \to \mathcal{Z}$ (defined by $M''(x) = M'(x, M(x))$) satisfies $(\rho_1 + \rho_2)$-zCDP.*

**Definition B.1** (Sensitivity). The sensitivity of a gradient query $\nabla_t$ to the dataset $\{x_i\}_{i=1}^N$ is

$$\Delta_2(\nabla_t) = \max_n \left\| \frac{1}{N} \sum_{j=1, j \neq n}^N \nabla_t^{(j)} - \frac{1}{N} \sum_{j=1}^N \nabla_t^{(j)} \right\|_2 = \frac{1}{N} \max_n \left\| \nabla_t^{(n)} \right\|_2 \qquad (19)$$

where $\nabla_t^{(n)}$ denotes the gradient of the $n$-th sample.

**Lemma B.2** (Gaussian mechanism (Bun & Steinke, 2016)). *Let $f : \mathcal{D}^n \to \mathcal{Z}$ have sensitivity $\Delta$. Define a randomized algorithm $M : \mathcal{D}^n \to \mathcal{Z}$ by $M(x) \gets f(x) + \mathcal{N}(0, \Delta^2 \sigma^2 I)$. Then $M$ satisfies $\frac{1}{2\sigma^2}$-zCDP.*

**Lemma B.3** ((Bun & Steinke, 2016)). *If $M$ is a mechanism satisfying $\rho$-zCDP, then $M$ is $(\rho + 2\sqrt{\rho \ln(1/\delta)}, \delta)$-DP for any $\delta > 0$.*

By solving $\rho + 2\sqrt{\rho \ln(1/\delta)} = \epsilon$, we can get $\rho = \epsilon + 2\ln(1/\delta) + 2\sqrt{\ln(1/\delta)(\epsilon + \ln(1/\delta))}$.

### B.2 Auxiliary lemmas

**Lemma B.4.** *If $\max_n \|x_n\|_2 = 1$ and $\frac{1}{N} \sum_n x_n = 0$, then the gradient sensitivity of the squared loss will be*

$$\Delta_2(\nabla) = \max_i \frac{1}{N} \sqrt{2f(\theta; x_i)} \|x_i\|_2 \leq \frac{1}{2}(DM \|\theta\|^2 + 1),$$

*where $\Theta_{\mathcal{M}}$ is the set of all possible parameters $\theta_t$ generated by the learning algorithm $\mathcal{M}$.*

*Proof.* According to the definition of sensitivity in Eq. (19), we have

$$\Delta_2(\nabla) = \max_i \left\| \nabla^{(i)} \right\|_2 = \max_n \frac{1}{n} \left\| A^{(i)} \theta - x_i \right\|_2$$

where we use $i$ denotes the index of sample in the dataset. Here, we assume it is constant 1. We may get

$$\left\| A^{(i)} \theta - x_i \right\|_2^2 = \left\| x_i(x_i^\top \theta - 1) \right\|_2^2 = (x_i^\top \theta - 1)^2 \|x_i\|_2^2 = 2f(\theta; x_i) \|x_i\|_2^2$$

where $f(\theta; x_i) = \frac{1}{2}(x_i^\top \theta - 1)^2$. Thus,

$$\Delta_2(\nabla) = \max_i \frac{1}{N} \sqrt{2f(\theta; x_i)} \|x_i\|_2$$

Since $\|x_n\|_2 \leq 1$ and $\frac{1}{N} \sum_{n=1}^N x_n = 0$,

$$f(\theta) = \frac{1}{2N} \sum_{n=1}^N [(x_n^\top \theta)^2 - 2x_n^\top \theta + 1] \leq \frac{1}{2N} \sum_{n=1}^N [(\|x_n\| \|\theta\|)^2 + 1] \leq \frac{1}{2}(DM \|\theta\|^2 + 1)$$

$\square$

**Lemma B.5.** *Assume assumptions in Theorem 4.4 are satisfied. Given variables defined in Theorem 4.4, the following inequality holds true:*

$$\sum_{t=1}^T \gamma^{T-t} \frac{2(1-\beta)\eta_t}{b_t} \sum_{i=1}^t \beta^{t-i} \|\nabla_t - \nabla_i\|^2 \leq \frac{\eta_0^3 \beta \gamma}{2M(1-\beta)^3(\gamma-\beta)^2} \sum_{i=1}^{T-1} \gamma^{T-i} \|v_{i+1}\|^2.$$

*Proof.* We first handle the inner summation. By smoothness, the inequality $\|\nabla f(x) - \nabla f(y)\| \leq M \|x - y\|$ holds true. Thus,

$$
\begin{aligned}
\sum\nolimits_{i=1}^{t} \beta^{t-i} \|\nabla_t - \nabla_i\|^2 &\leq M^2 \sum\nolimits_{i=1}^{t} \beta^{t-i} \|\theta_t - \theta_i\|^2 \\
&= M^2 \sum\nolimits_{k=0}^{t-1} \beta^k \|\theta_t - \theta_{t-k}\|^2 \\
&= M^2 \sum\nolimits_{k=0}^{t-1} \beta^k \left\| \sum\nolimits_{i=t-k}^{t-1} \eta_i v_{i+1}/b_i \right\|^2 \\
&\leq M^2 \sum\nolimits_{k=0}^{t-1} \beta^k \left( \sum\nolimits_{j=t-k}^{t-1} \eta_j^2/b_j^2 \right) \left( \sum\nolimits_{i=t-k}^{t-1} \|v_{i+1}\|^2 \right)
\end{aligned}
$$

where the last inequality is by Cauchy-Schwartz inequality. Because $\frac{1}{b_t} = \frac{1}{1-\beta^t} \leq \frac{1}{1-\beta}$ and $\eta_t = \frac{\eta_0}{2M}$,

$$
\begin{aligned}
\sum\nolimits_{i=1}^{t} \beta^{t-i} \|\nabla_t - \nabla_i\|^2 &\leq \frac{\eta_0^2}{4(1-\beta)^2} \sum\nolimits_{k=0}^{t-1} \beta^k k \sum\nolimits_{i=t-k}^{t-1} \|v_{i+1}\|^2 \\
&= \frac{\eta_0^2}{4(1-\beta)^2} \sum\nolimits_{k=0}^{t-1} \beta^k k \sum\nolimits_{i=1}^{t-1} \|v_{i+1}\|^2 \, \mathbb{I}(i \geq t-k) \\
&= \frac{\eta_0^2}{4(1-\beta)^2} \sum\nolimits_{i=1}^{t-1} \|v_{i+1}\|^2 \sum\nolimits_{k=0}^{t-1} \beta^k k \, \mathbb{I}(k \geq t-i) \\
&= \frac{\eta_0^2}{4(1-\beta)^2} \sum\nolimits_{i=1}^{t-1} \|v_{i+1}\|^2 \sum\nolimits_{k=t-i}^{t-1} \beta^k k \qquad (20)
\end{aligned}
$$

where $\mathbb{I}(\cdot)$ is the indicating function which output 1 if the condition holds true, otherwise 0.

Denote the left-hand-side of the conclusion as LHS. We plug Eq. (20) into LHS to get

$$
\begin{aligned}
\text{LHS} &\leq \sum_{t=1}^{T} \gamma^{T-t} \frac{1}{b_t} \frac{\eta_0^3}{4M(1-\beta)} \sum\nolimits_{i=1}^{t-1} \|v_{i+1}\|^2 \sum\nolimits_{k=t-i}^{t-1} \beta^k k \\
&\leq \frac{\eta_0^3}{4M(1-\beta)^2} \sum_{t=1}^{T} \gamma^{T-t} \sum\nolimits_{i=1}^{t-1} \|v_{i+1}\|^2 \sum\nolimits_{k=t-i}^{t-1} \beta^k k
\end{aligned}
$$

where we relax the upper bound by $\frac{1}{b_t} = \frac{1}{1-\beta^t} \leq \frac{1}{1-\beta}$. Using Lemma B.6 can directly lead to the conclusion:

$$
\text{LHS} \leq \frac{\eta_0^3 \beta \gamma}{2M(1-\beta)^3(\gamma-\beta)^2} \sum_{i=1}^{T-1} \gamma^{T-i} \|v_{i+1}\|^2 .
$$

$\square$

**Lemma B.6.** *Given variables defined in Theorem 4.4, the following inequality holds true:*

$$
\sum_{t=1}^{T} \gamma^{T-t} \sum\nolimits_{i=1}^{t-1} \|v_{i+1}\|^2 \sum\nolimits_{k=t-i}^{t-1} k\beta^k \leq \frac{2\beta\gamma}{(\gamma-\beta)^2(1-\beta)} \sum_{i=1}^{T-1} \gamma^{T-i} \|v_{i+1}\|^2 .
$$

*Proof.* We first derive the summation:

$$
\begin{aligned}
U_1(t,i) &\triangleq \sum_{k=t-i}^{t-1} \beta^k k = \sum_{k=t-i}^{t-1} \sum_{j=1}^{k} \beta^k \\
&= \sum_{k=t-i}^{t-1} \sum_{j=1}^{t-1} \beta^k \mathbb{I}(j \leq k) \\
&= \sum_{j=1}^{t-1} \sum_{k=\max(t-i,j)}^{t-1} \beta^k \\
&= \sum_{j=1}^{t-1} \frac{\beta^{\max(t-i,j)} - \beta^t}{1-\beta} \\
&= \frac{1}{1-\beta}\left( (t-i)\beta^{t-i} + \frac{\beta^{t-i+1} - \beta^t}{1-\beta} - \frac{\beta - \beta^t}{1-\beta} \right) \\
&= \frac{1}{1-\beta}\left( (t-i)\beta^{t-i} + \frac{\beta^{t-i+1} - \beta}{1-\beta} \right)
\end{aligned}
$$

Now, we substitute $U_1(t,i)$ into LHS and replace $t-i$ by $j$, i.e., $t = j+i$, to get

$$
\begin{aligned}
\text{LHS} &= \sum_{t=1}^{T} \gamma^{T-t} \sum_{i=1}^{t-1} \|v_{i+1}\|^2 \frac{1}{1-\beta}\left( (t-i)\beta^{t-i} + \frac{\beta^{t-i+1} - \beta}{1-\beta} \right) \\
&= \sum_{i=1}^{T-1} \|v_{i+1}\|^2 \sum_{t=i+1}^{T} \gamma^{T-t} \frac{1}{1-\beta}\left( (t-i)\beta^{t-i} + \frac{\beta^{t-i+1} - \beta}{1-\beta} \right) \\
&= \sum_{i=1}^{T-1} \|v_{i+1}\|^2 \sum_{j=1}^{T-i} \gamma^{T-(j+i)} \frac{1}{1-\beta}\left( j\beta^{j} + \frac{\beta^{j+1} - \beta}{1-\beta} \right) \\
&= \sum_{i=1}^{T-1} \gamma^{T-i} \|v_{i+1}\|^2 \sum_{j=1}^{T-i} \gamma^{-j} \frac{1}{1-\beta}\left( j\beta^{j} + \frac{\beta^{j+1} - \beta}{1-\beta} \right) \\
&\leq \frac{1}{1-\beta} \sum_{i=1}^{T-1} \gamma^{T-i} \|v_{i+1}\|^2 \sum_{j=1}^{T-i} \left( j\left(\frac{\beta}{\gamma}\right)^{j} + \frac{\beta}{1-\beta}\left(\frac{\beta}{\gamma}\right)^{j} \right)
\end{aligned}
$$

Let $a = \beta/\gamma$, we show

$$
\begin{aligned}
\sum_{j=1}^{T-i} j a^{j} &= \sum_{j=1}^{T-i} \sum_{o=1}^{j} a^{j} \\
&= \sum_{o=1}^{T-i} \sum_{j=o}^{T-i} a^{j} \\
&= \sum_{o=1}^{T-i} \left( \frac{a^{o} - a^{T-i+1}}{1-a} \right) \\
&= \frac{a - a^{T-i+1}}{(1-a)^2} - (T-i)\frac{a^{T-i+1}}{1-a} \\
&\leq \frac{a}{(1-a)^2}.
\end{aligned}
$$

Thus,

$$\text{LHS} \leq \frac{1}{1-\beta} \sum_{i=1}^{T-1} \gamma^{T-i} \|v_{i+1}\|^2 \left( \frac{a}{(1-a)^2} + \frac{\beta}{1-\beta} \sum_{j=1}^{T-i} a^j \right)$$

$$\leq \frac{1}{1-\beta} \sum_{i=1}^{T-1} \gamma^{T-i} \|v_{i+1}\|^2 \left( \frac{a}{(1-a)^2} + \frac{\beta}{1-\beta} \frac{a}{1-a} \right)$$

$$\leq \frac{a}{(1-a)^2(1-\beta)} \sum_{i=1}^{T-1} \gamma^{T-i} \|v_{i+1}\|^2$$

Because $\gamma < 1$, $\beta < a = \beta/\gamma$ and

$$\frac{a}{(1-a)^2} + \frac{\beta}{1-\beta} \frac{a}{1-a} \leq \frac{2a}{(1-a)^2}.$$

Therefore,

$$\text{LHS} \leq \frac{2a}{(1-a)^2(1-\beta)} \sum_{i=1}^{T-1} \gamma^{T-i} \|v_{i+1}\|^2 = \frac{2\beta\gamma}{(\gamma-\beta)^2(1-\beta)} \sum_{i=1}^{T-1} \gamma^{T-i} \|v_{i+1}\|^2$$

$\square$

**Lemma B.7.** *Suppose $\gamma \in (0,1)$ and $\beta \in (0,1)$. Define*

$$\hat{T} = \max t \text{ s.t. } \gamma^{t-1} \geq \frac{1-\beta}{1-\beta^t}.$$

*If $t \leq \hat{T}$, $\frac{1-\beta}{1-\beta^t} \leq \gamma^{t-1}$ for $t = 1, \ldots, T$. If $t > \hat{T}$, $\frac{1-\beta}{1-\beta^t} < \gamma^{\hat{T}-1}$.*

*Proof.* Define $h(t) = \gamma^{t-1}(1-\beta^t)$ whose derivatives are

$$h'(t) = \gamma^{t-1}(1-\beta^t)\ln\gamma + \gamma^{t-1}(-\beta^t)\ln\beta$$
$$= \gamma^{t-1}\left[\ln\gamma - \beta^t(\ln\gamma + \ln\beta)\right]$$
$$= \gamma^{t-1}\left[1 - \beta^t(1 + \log_\gamma\beta)\right]\ln\gamma.$$

Simple calculation shows $1 - \beta^t(1 + \log_\gamma\beta)\big|_{t=0} = -\log_\gamma\beta < 0$ and $\lim_{t\to+\infty} 1 - \beta^t(1 + \log_\gamma\beta) = 1$. When $t = -\log_\beta(1 + \log_\gamma\beta)$ denoted as $t_0$, $1 - \beta^t(1 + \log_\gamma\beta) = 0$. Because $1 - \beta^t(1 + \log_\gamma\beta)$ is monotonically increasing by $t$ and $\gamma^{t-1}\ln\gamma$ is negative, $h'(t) \geq 0$ if $t \leq t_0$. Otherwise, $h'(t) < 0$. Therefore, $h(t)$ is a concave function. Because $h(1) = 1 - \beta$ and $h(\hat{T}) = \gamma^{\hat{T}-1}(1-\beta^{\hat{T}}) \geq 1 - \beta > 0$, $h(t) \geq 1 - \beta$ for $t = 1, \ldots, \hat{T}$. Thus, for all $t \in [1, \hat{T}]$, we have $\frac{1-\beta}{1-\beta^t} \leq \gamma^{t-1}$.

For $t > \hat{T}$, because $\frac{1-\beta}{1-\beta^t}$ monotonically increases by $t$, we have $\frac{1-\beta}{1-\beta^t} < \frac{1-\beta}{1-\beta^{\hat{T}}} \leq \gamma^{\hat{T}-1}$. $\square$

## C PROOFS

*Proof of Theorem 3.1.* Because all sample gradient are $G$-Lipschitz continuous, the sensitivity of the averaged gradient is upper bounded by $G/N$. Based on Lemma B.2, the privacy cost of $g_t$ is $\frac{1}{2\sigma_t^2}$[1].

Here, we make the output of each iteration a tuple of $(\theta_{t+1}, v_{t=1})$. For the 1st iteration, because $\theta_1$ does not embrace private information by random initialization, the mapping,

$$\begin{bmatrix} v_2 \\ \theta_2 \end{bmatrix} = \begin{bmatrix} g_1 \\ \theta_1 - \eta_1 g_1 \end{bmatrix},$$

---

[1]For brevity, when we say the privacy cost of some value, e.g., gradient, we actually refer to the cost of mechanism that output the value.

is $\hat{\rho}_1$-zCDP where $\hat{\rho}_1 = \frac{1}{2\sigma_t^2}$.

Suppose the output of the $t$-th iteration, $(\theta_t, v_t)$, is $\hat{\rho}_t$-zCDP. At each iteration, we have the following mapping $(\theta_t, v_t) \rightarrow (\theta_{t+1}, v_{t+1})$ defined as

$$
\left[ \begin{array}{c} v_{t+1} \\ \theta_{t+1} \end{array} \right] = \left[ \begin{array}{c} \phi(v_t, g_t) \\ \theta_t - \eta_t \phi(v_t, g_t) \end{array} \right].
$$

Thus, the output tuple $(\theta_{t+1}, v_{t+1})$ is $(\hat{\rho}_t + \frac{1}{2\sigma_t^2})$-zCDP by Lemma B.1.

Thus, the recursion implies that $(\theta_{T+1}, v_{T+1})$ has privacy cost as

$$
\hat{\rho}_{T+1} = \hat{\rho}_T + \frac{1}{2\sigma_T^2} = \cdots = \sum_{t=1}^{T} \frac{1}{2\sigma_t^2} = \frac{1}{2}\sum_{t=1}^{T} \rho_t \leq \frac{1}{2}(R - R_T) \leq \frac{1}{2}R.
$$

Let $\rho = \hat{\rho}_{T+1}$. Then we can get the conclusion. $\qquad \square$

## C.1 GRADIENT DESCENTS

*Proof of Theorem 4.1.* With the definition of smoothness in Definition 3.3 and Eq. (1), we have

$$
\begin{aligned}
f(\theta_{t+1}) - f(\theta_t) &\leq -\eta_t \nabla_t^\top (\nabla_t + G\sigma_t \nu_t/N) + \frac{1}{2}M\eta_t^2 \|\nabla_t + G\sigma_t \nu_t/N\|^2 \\
&= -\eta_t(1 - \frac{1}{2}M\eta_t)\|\nabla_t\|^2 - (1 - M\eta_t)\eta_t \nabla_t^\top G\sigma_t \nu_t/N + \frac{1}{2}M\eta_t^2 \|G\sigma_t \nu_t/N\|^2 \\
&\leq -2\mu\eta_t(1 - \frac{1}{2}M\eta_t)(f(\theta_t) - f(\theta^*)) - (1 - M\eta_t)\eta_t \nabla_t^\top G\sigma_t \nu_t/N \\
&\quad + \frac{1}{2}M\eta_t^2 \|G\sigma_t \nu_t/N\|^2.
\end{aligned}
$$

where the last inequality is due to the Polyak-Lojasiewicz condition. Taking expectation on both sides, we can obtain

$$
\mathbb{E}[f(\theta_{t+1})] - \mathbb{E}[f(\theta_t)] \leq -2\mu\eta_t(1 - \frac{M}{2}\eta_t)(\mathbb{E}[f(\theta_t)] - f(\theta^*)) + \frac{M}{2}(\eta_t G\sigma_t/N)^2 \mathbb{E}\|\nu_t\|^2
$$

which can be reformulated by substacting $f(\theta^*)$ on both sides and re-arranged as

$$
\mathbb{E}[f(\theta_{t+1})] - f(\theta^*) \leq \left(1 - 2\mu\eta_t(1 - \frac{M}{2}\eta_t)\right)(\mathbb{E}[f(\theta_t)] - f(\theta^*)) + \frac{M}{2}(\eta_t G\sigma_t/N)^2 D
$$

Recursively using the inequality, we can get

$$
\begin{aligned}
\mathbb{E}[f(\theta_{T+1})] - f(\theta^*) &\leq \prod_{t=1}^{T}\left(1 - 2\mu\eta_t(1 - \frac{M}{2}\eta_t)\right)(\mathbb{E}[f(\theta_1)] - f(\theta^*)) \\
&\quad + \frac{MD}{2}\sum_{t=1}^{T}\prod_{i=t+1}^{T}\left(1 - 2\mu\eta_i(1 - \frac{M}{2}\eta_i)\right)(\eta_t G\sigma_t/N)^2.
\end{aligned}
$$

Let $\eta_t \equiv 1/M$. Then the above inequality can be simplified as

$$
\begin{aligned}
\mathbb{E}[f(\theta_{T+1})] - f(\theta^*) &\leq \gamma^T(\mathbb{E}[f(\theta_1)] - f(\theta^*)) + R\sum_{t=1}^{T}\gamma^{T-t}\frac{MD}{2R}\left(\frac{\eta_t G}{N}\right)^2 \sigma_t^2 \\
&= \gamma^T(\mathbb{E}[f(\theta_1)] - f(\theta^*)) + R\sum_{t=1}^{T}\gamma^{T-t}\alpha\sigma_t^2(\mathbb{E}[f(\theta_1)] - f(\theta^*)) \\
&= \left(\gamma^T + R\sum_{t=1}^{T}q_t\sigma_t^2\right)(f(\theta_1) - f(\theta^*))
\end{aligned}
$$

$\qquad \square$

*Proof of Theorem 4.2.* The minimizer of the upper bound of Eq. (3) can be written as

$$T^* = \arg\min_T \gamma^T + \alpha\kappa(1 - \gamma^T)T \tag{21}$$

where we substitute $\sigma^2 = T/R$ in the second line. To find the convex minimization problem, we need to vanishing its gradient which involves an equation like $T\gamma^T = c$ for some real constant $c$. However, the solution is $W_k(c)$ for some integer $k$ where $W$ is Lambert W function which does not have a simple analytical form. Instead, because $\gamma^T > 0$, we can minimize a surrogate upper bound as following

$$T^* = \arg\min_T \gamma^T + \alpha\kappa T = \frac{1}{\ln(1/\gamma)} \ln\left(\frac{\ln(1/\gamma)}{\kappa\alpha}\right), \text{ if } \kappa\alpha + \ln\gamma < 0 \tag{22}$$

where we use the surrogate upper bound in the second line and utilize $\gamma = 1 - \frac{1}{\kappa}$. However, the minimizer of the surrogate objective is not optimal for the original objective. When $\kappa$ is large, the term, $-\alpha\kappa\gamma^T T$, cannot be neglected as we expect. On the other hands, $T$ suffers from explosion if $\kappa \to \infty$ and meanwhile $1/\gamma \to_+ 1$. The tendency is counterintuitive since a small $T$ should be taken for sharp losses. To fix the issue, we change the form of $T^*$ as

$$T^* = \frac{1}{\ln(1/\gamma)} \ln\left(1 + \frac{\ln(1/\gamma)}{\alpha}\right), \tag{23}$$

which gradually converges to 0 as $\kappa \to \infty$.

Now we substitute Eq. (23) into the original objective function, Eq. (21), to get

$$\text{ERUB}^{\text{uniform}} = \frac{1}{1 + \ln(1/\gamma)/\alpha} \left[1 + \kappa\ln\left(1 + \frac{\ln(1/\gamma)}{\alpha}\right)\right]. \tag{24}$$

Notice that

$$\ln(1/\gamma) = \ln(\kappa/(\kappa - 1)) = \ln(1 + 1/(\kappa - 1)) \leq \frac{1}{\kappa - 1} \leq \frac{1}{c\kappa}$$

because $\kappa \geq \frac{1}{1-c} > 1$ for some constant $c \in (0, 1)$. In addition,

$$\ln(1/\gamma) = -\ln(1 - 1/\kappa) \geq 1/\kappa.$$

Now, we can get the upper bound of Eq. (24) as

$$\text{ERUB}^{\text{uniform}} \leq \frac{\kappa}{\kappa + 1/\alpha}\left[1 + \kappa\ln\left(1 + \frac{1}{c\kappa\alpha}\right)\right]$$

$$\leq c_1 \frac{\kappa}{\kappa + 1/\alpha}\kappa\left[\ln\left(1 + \frac{1}{\kappa\alpha}\right) + \ln(\frac{1}{c}))\right]$$

$$\leq c_1 c_2 \frac{\kappa^2}{\kappa + 1/\alpha}\ln\left(1 + \frac{1}{\kappa\alpha}\right)$$

for some constants $c_1, c_2$ and large enough $\frac{1}{\alpha}$. Also, we can get the lower bound

$$\text{ERUB}^{\text{uniform}} \geq \frac{c\kappa}{c\kappa + 1/\alpha}\left[1 + \kappa\ln\left(1 + \frac{1}{\kappa\alpha}\right)\right] \geq c\frac{\kappa^2}{\kappa + 1/\alpha}\ln\left(1 + \frac{1}{\kappa\alpha}\right).$$

where we use the condition $c \in (0, 1)$. Thus, $\text{ERUB}^{\text{uniform}} = \Theta\left(\frac{\kappa^2}{\kappa + 1/\alpha}\ln\left(1 + \frac{1}{\kappa\alpha}\right)\right)$. □

*Proof of Lemma 4.1.* By $\sum_{t=1}^{T} \sigma^{-2} = R$ and Cauchy-Schwarz inequality, we can derive the achievable lower bound as

$$R\sum_t q_t\sigma_t^2 = \sum_t \frac{1}{\sigma_t^2}\sum_t q_t\sigma_t^2 \geq \left(\sum_{t=1}^{T}\sqrt{q_t}\right)^2$$

where the inequality becomes equality if and only if $s/\sigma_t^2 = q_t \sigma_t^2$, i.e., $\sigma_t = (s/q_t)^{1/4}$, for some positive constant $s$. The equality $\sum_{t=1}^{T} \sigma_t^{-2} = R$ immediately suggests $\sqrt{s} = \frac{1}{R} \sum_{t=1}^{T} \sqrt{q_t}$. Thus, we get the $\sigma_t$.

Notice

$$T \sum_{t=1}^{T} q_t - \left( \sum_{t=1}^{T} \sqrt{q_t} \right)^2 = T^2 \frac{1}{T} \sum_{t=1}^{T} \left( \sqrt{q_t} - \frac{1}{T} \sum_{i=1}^{T} \sqrt{q_i} \right)^2 = T^2 \operatorname{Var}[q_t] \qquad (25)$$

where the variance is w.r.t. $t$. $\qquad \square$

*Proof of Theorem 4.3.* The upper bound of Eq. (3) can be written as

$$\mathrm{ERUB}^{\mathrm{dyn}} = \gamma^T + \sum_{t=1}^{T} \gamma^{T-t} \alpha R \sigma^2$$
$$= \gamma^T + \alpha \left( \sum_{t=1}^{T} \sqrt{\gamma^{T-t}} \right)^2$$
$$= \gamma^T + \alpha \left( \frac{1 - \gamma^{T/2}}{1 - \sqrt{\gamma}} \right)^2$$

where we make use of Lemma 4.1. Then, the minimizer of the ERUB is

$$T^* = \arg\min_{T} \gamma^T + \alpha \left( \frac{1 - \gamma^{T/2}}{1 - \sqrt{\gamma}} \right)^2$$
$$= 2 \log_\gamma \left( \frac{\alpha}{\alpha + (1 - \sqrt{\gamma})^2} \right). \qquad (26)$$

We can substitute Eq. (26) into $\mathrm{ERUB}^{\mathrm{dyn}}$ to get

$$\mathrm{ERUB}^{\mathrm{dyn}}_{\mathrm{min}} = \left( \frac{\alpha}{\alpha + (1 - \sqrt{\gamma})^2} \right)^2 + \alpha \left( \frac{1}{1 - \sqrt{\gamma}} \right)^2 \left( 1 - \frac{\alpha}{\alpha + (1 - \sqrt{\gamma})^2} \right)^2$$
$$= \left( \frac{\alpha(1 - \sqrt{\gamma})^{-2}}{\alpha(1 - \sqrt{\gamma})^{-2} + 1} \right)^2 + \frac{\alpha(1 - \sqrt{\gamma})^{-2}}{\left( \alpha(1 - \sqrt{\gamma})^{-2} + 1 \right)^2}$$
$$= \frac{\alpha(1 - \sqrt{\gamma})^{-2}}{\alpha(1 - \sqrt{\gamma})^{-2} + 1}$$

Notice that $\left(1 - \sqrt{\gamma}\right)^{-2} = \kappa^2 + \kappa^2 - \kappa + 2\kappa\sqrt{\kappa(\kappa - 1)} = \kappa(2\kappa - 1 + 2\sqrt{\kappa(\kappa - 1)})$ and it is bounded by

$$\kappa(2\kappa - 1 + 2\sqrt{\kappa(\kappa - 1)}) \leq 4\kappa^2,$$
$$\kappa(2\kappa - 1 + 2\sqrt{\kappa(\kappa - 1)}) \geq \kappa(2\kappa - (3\kappa - 2) + 2\sqrt{(\kappa - 1)(\kappa - 1)}) = \kappa(-\kappa + 2 + 2\kappa - 2) = \kappa^2.$$

Therefore, $\kappa \leq \left(1 - \sqrt{\gamma}\right)^{-1} \leq 2\kappa$, with which we can derive

$$\mathrm{ERUB}^{\mathrm{dyn}}_{\mathrm{min}} \leq 4 \frac{\kappa^2 \alpha}{\kappa^2 \alpha + 1},$$
$$\mathrm{ERUB}^{\mathrm{dyn}}_{\mathrm{min}} \geq \frac{\kappa^2 \alpha}{4\kappa^2 \alpha + 1} \geq \frac{1}{4} \frac{\kappa^2 \alpha}{\kappa^2 \alpha + 1}.$$

Thus, $\mathrm{ERUB}^{\mathrm{dyn}}_{\mathrm{min}} = \Theta\left( \frac{\kappa^2 \alpha}{\kappa^2 \alpha + 1} \right)$. $\qquad \square$

## C.2 GRADIENT DESCENTS WITH MOMENTUM

*Proof of Theorem 4.4.* Without loss of generality, we absorb the $C\sigma_t/N$ into the variance of $\nu_t$ such that $\nu_t \sim \mathcal{N}(0, \frac{C\sigma_t^2}{N}I)$ and $g_t \leftarrow \nabla_t + \nu_t$. Define $b_t = 1 - \beta^t$.

By smoothness and Eq. (1), we have

$$
\begin{aligned}
f(\theta_{t+1}) - f(\theta_t) &\leq \nabla_t^\top (\theta_{t+1} - \theta_t) + \frac{1}{2} M \|\theta_{t+1} - \theta_t\|^2 \\
&= -\frac{\eta_t}{b_t^2} b_t \nabla_t^\top v_{t+1} + \frac{1}{2} M \frac{\eta_t^2}{b_t^2} \|v_{t+1}\|^2 \\
&= \frac{\eta_t}{b_t^2} \left( \|b_t \nabla_t - v_{t+1}\|^2 - \|b_t \nabla_t\|^2 - \|v_{t+1}\|^2 \right) + \frac{1}{2} M \frac{\eta_t^2}{b_t^2} \|v_{t+1}\|^2 \\
&= \frac{\eta_t}{b_t^2} \underbrace{\|b_t \nabla_t - v_{t+1}\|^2}_{U_1(t)} - \eta_t \|\nabla_t\|^2 - \frac{\eta_t}{b_t^2} (1 - \frac{1}{2} M \eta_t) \|v_{t+1}\|^2,
\end{aligned} \tag{27}
$$

where only the $U_1(t)$ is non-negative. Specifically, $U_1(t)$ describes the difference between current gradient and the average. We can expand $v_{t+1}$ to get an upper bound:

$$
\begin{aligned}
U_1(t) &= \|b_t \nabla_t - v_{t+1}\|^2 \\
&= \left\| (1-\beta) \sum_{i=1}^t \beta^{t-i} \nabla_t - (1-\beta) \sum_{i=1}^t \beta^{t-i} g_i \right\|^2 \\
&= (1-\beta)^2 \left\| \sum_{i=1}^t \beta^{t-i} (\nabla_t - g_i) \right\|^2 \\
&= (1-\beta)^2 \left\| \sum_{i=1}^t \beta^{t-i} (\nabla_t - \nabla_i) + \sum_{i=1}^t \beta^{t-i} (\nabla_i - g_i) \right\|^2 \\
&\leq 2(1-\beta)^2 \left[ \left\| \sum_{i=1}^t \beta^{t-i} (\nabla_t - \nabla_i) \right\|^2 + \left\| \sum_{i=1}^t \beta^{t-i} (\nabla_i - g_i) \right\|^2 \right] \\
&\leq 2(1-\beta) \left[ b_t \underbrace{\sum_{i=1}^t \beta^{t-i} \|\nabla_t - \nabla_i\|^2}_{U_2(t) \text{ (gradient variance)}} + (1-\beta) \underbrace{\left\| \sum_{i=1}^t \beta^{t-i} \nu_i \right\|^2}_{\text{noise variance}} \right]
\end{aligned}
$$

where we use $\|x + y\|^2 \leq (\|x\| + \|y\|)^2 \leq 2(\|x\|^2 + \|y\|^2)$. The last inequality can be proved by Cauchy-Schwartz inequality for each coordinate.

We plug the $U_1(t)$ into Eq. (27) and use the PL condition to get

$$
\begin{aligned}
f(\theta_{t+1}) - f(\theta_t) &\leq \frac{\eta_t}{b_t^2} U_1(t) - \eta_t \|\nabla_t\|^2 - \frac{\eta_t}{b_t^2} (1 - \frac{1}{2} M \eta_t) \|v_{t+1}\|^2 \\
&\leq -\eta_t \|\nabla_t\|^2 + \frac{\eta_t}{b_t^2} 2(1-\beta) \left[ b_t U_2(t) + (1-\beta) \left\| \sum_{i=1}^t \beta^{t-i} \nu_i \right\|^2 \right] \\
&\quad - \frac{\eta_t}{b_t^2} (1 - \frac{1}{2} M \eta_t) \|v_{t+1}\|^2 \\
&\leq -2\mu\eta_t (f(\theta_t) - f(\theta^*)) + \frac{2(1-\beta)\eta_t}{b_t} U_2(t) + \frac{2(1-\beta)^2 \eta_t}{b_t^2} \left\| \sum_{i=1}^t \beta^{t-i} \nu_i \right\|^2 \\
&\quad - \frac{\eta_t}{b_t^2} (1 - \frac{1}{2} M \eta_t) \|v_{t+1}\|^2.
\end{aligned}
$$

Rearranging terms and taking expectation to show

$$\mathbb{E}[f(\theta_{t+1})] - f(\theta^*) \leq \gamma(\mathbb{E}[f(\theta_t)] - f(\theta^*)) + \frac{2(1-\beta)^2\eta_t}{b_t^2}\sum_{i=1}^{t}\beta^{t-i}\mathbb{E}\|\nu_i\|^2$$

$$+ \frac{2(1-\beta)\eta_t}{b_t}\mathbb{E}[U_2(t)] - \frac{\eta_t}{b_t^2}(1 - \frac{1}{2}M\eta_t)\mathbb{E}\|v_{t+1}\|^2$$

$$= \gamma(\mathbb{E}[f(\theta_t)] - f(\theta^*)) + \frac{2(1-\beta)^2\eta_t}{b_t^2}\sum_{i=1}^{t}\beta^{2(t-i)}\frac{C^2D\sigma_t^2}{N^2}$$

$$+ \frac{2(1-\beta)\eta_t}{b_t}\mathbb{E}[U_2(t)] - \frac{\eta_t}{b_t^2}(1 - \frac{1}{2}M\eta_t)\mathbb{E}\|v_{t+1}\|^2$$

where $\gamma = 1 - \eta_0/\kappa = 1 - 2\mu\eta_t$. The recursive inequality implies

$$\mathbb{E}[f(\theta_{T+1})] - f(\theta^*) \leq \gamma^T(f(\theta_1) - f(\theta^*)) + \sum_{t=1}^{T}\gamma^{T-t}\frac{2(1-\beta)^2\eta_t}{b_t^2}\sum_{i=1}^{t}\beta^{2(t-i)}\frac{C^2D\sigma_t^2}{N^2}$$

$$+ \sum_{t=1}^{T}\gamma^{T-t}\frac{2(1-\beta)\eta_t}{b_t}\mathbb{E}[U_2(t)] - \sum_{t=1}^{T}\gamma^{T-t}\frac{\eta_t}{b_t^2}(1 - \frac{1}{2}M\eta_t)\mathbb{E}\|v_{t+1}\|^2$$

$$= \left(\gamma^T + 2\eta_0\alpha R\underbrace{\sum_{t=1}^{T}\gamma^{T-t}\frac{(1-\beta)^2}{b_t^2}\sum_{i=1}^{t}\beta^{2(t-i)}\sigma_t^2}_{U_3}\right)(f(\theta_1) - f(\theta^*))$$

$$+ \underbrace{\sum_{t=1}^{T}\gamma^{T-t}\frac{2(1-\beta)\eta_t}{b_t}\mathbb{E}[U_2(t)] - \sum_{t=1}^{T}\gamma^{T-t}\frac{\eta_t}{b_t^2}(1 - \frac{1}{2}M\eta_t)\mathbb{E}\|v_{t+1}\|^2}_{U_4(t)}.$$

where we utilize $\alpha = \frac{DC^2}{2MN^2R}\frac{1}{f(\theta_1)-f(\theta^*)}$ and $\eta_t = \frac{\eta_0}{2M}$.

By Lemma B.5, we have

$$\sum_{t=1}^{T}\gamma^{T-t}\frac{2(1-\beta)\eta_t}{b_t}U_2(t) \leq \frac{\eta_0^3\beta\gamma}{2M(1-\beta)^3(\gamma-\beta)^2}\sum_{i=1}^{T-1}\gamma^{T-i}\|v_{i+1}\|^2.$$

Thus, by $\frac{1}{b_t} \geq 1$,

$$U_4(t) \leq \frac{\eta_0^3\beta\gamma}{2M(1-\beta)^3(\gamma-\beta)^2}\sum_{i=1}^{T-1}\gamma^{T-i}\mathbb{E}\|v_{i+1}\|^2 - \frac{\eta_0}{2M}(1 - \frac{\eta_0}{4})\sum_{t=1}^{T}\gamma^{T-t}\mathbb{E}\|v_{t+1}\|^2$$

$$= -\frac{\eta_0}{2M}\zeta\sum_{t=1}^{T}\gamma^{T-t}\mathbb{E}\|v_{t+1}\|^2$$

where

$$\zeta = 1 - \frac{1}{4}\eta_0 - \frac{\beta\gamma}{(\gamma-\beta)^2(1-\beta)^3}\eta_0^2 = 1 - \frac{1}{4}\eta_0 - \frac{\beta/\gamma}{(1-\beta/\gamma)^2(1-\beta)^3}\eta_0^2$$

When a small enough $\eta_0$, e.g., Specifically,

$$\eta_0 \leq \frac{(\gamma-\beta)^2(1-\beta)^3}{8\beta\gamma}\left[\sqrt{1 + \frac{64\beta\gamma}{(\gamma-\beta)^2(1-\beta)^3}} - 1\right]$$

$$= \frac{8}{\sqrt{1 + 64\beta\gamma(\gamma-\beta)^{-2}(1-\beta)^{-3}} + 1}$$

We can have $\zeta \geq 0$.

By the definition of $U_3(T,\sigma)$, we can get

$$\mathbb{E}[f(\theta_{T+1})] - f(\theta^*) \leq \left(\gamma^T + 2\eta_0\alpha RU_3(T,\sigma)\right)(f(\theta_1) - f(\theta^*)) - \frac{\eta_0}{2M}\zeta\sum_{t=1}^{T}\gamma^{T-t}\mathbb{E}\|v_{t+1}\|^2.$$

$\square$

*Proof of Theorem 4.5.* Since $\sigma_t$ is static, by definition of $U_3$ in Theorem 4.4,

$$
\begin{aligned}
U_3 &= \sum_{t=1}^{T} \gamma^{T-t} \frac{(1-\beta)^2}{(1-\beta^t)^2} \sum_{i=1}^{t} \beta^{2(t-i)} \sigma^2 \\
&= \sigma^2 \sum_{t=1}^{T} \gamma^{T-t} \frac{(1-\beta)^2}{(1-\beta^t)^2} \sum_{i=1}^{t} \beta^{2(t-i)} \\
&= \sigma^2 \sum_{t=1}^{T} \gamma^{T-t} \frac{(1-\beta)^2}{(1-\beta^t)^2} \frac{1-\beta^{2t}}{1-\beta^2} \\
&= \sigma^2 \sum_{t=1}^{T} \gamma^{T-t} \frac{1-\beta}{1-\beta^t} \frac{1+\beta^t}{1+\beta}.
\end{aligned}
$$

Because $\frac{1-\beta}{1-\beta^t}\frac{1+\beta^t}{1+\beta} \le 1$, the $U_3$ will be smaller than the corresponding summation in GD with uniform schedule.

By Lemma B.7, when $T > \hat{T}$, we can rewrite $U_3$ as

$$
\begin{aligned}
U_3 &\le \sigma^2 \sum_{t=1}^{T} \gamma^{T-t} \frac{1-\beta}{1-\beta^t} \\
&= \sigma^2 \sum_{t=1}^{\hat{T}} \gamma^{T-t} \frac{1-\beta}{1-\beta^t} + \sigma^2 \sum_{t=\hat{T}+1}^{T} \gamma^{T-t} \frac{1-\beta}{1-\beta^t} \\
&\le \sigma^2 \sum_{t=1}^{\hat{T}} \gamma^{T-t} \gamma^{t-1} + \sigma^2 \sum_{t=\hat{T}+1}^{T} \gamma^{T-t} \gamma^{\hat{T}-1} \\
&= \sigma^2 \gamma^{T-1} \hat{T} + \sigma^2 \gamma^{\hat{T}-1} \sum_{t=1}^{T-\hat{T}} \gamma^{T-\hat{T}-t} \\
&= \sigma^2 \gamma^{T-1} \hat{T} + \sigma^2 \frac{\gamma^{\hat{T}-1} - \gamma^{T-1}}{1-\gamma} \\
&= \frac{T}{\gamma R} \gamma^T \left( \hat{T} + \frac{\gamma^{\hat{T}-T} - 1}{1-\gamma} \right)
\end{aligned}
$$

where we use $\sigma^2 = T/R$ in the last line. Without assuming $T > \hat{T}$, we can generally write the upper bound as

$$
U_3 \le \frac{T}{\gamma R} \gamma^T \left( \min\{\hat{T}, T\} + \max\{\frac{\gamma^{\hat{T}-T}-1}{1-\gamma}, 0\} \right).
$$

By Theorem 4.4, because $\zeta \ge 0$, we have

$$
\begin{aligned}
\text{ERUB} &\le \gamma^T + 2R\eta_0\alpha U_3 \\
&= \gamma^T (1 + \frac{\alpha'}{\gamma} T \left( \min\{\hat{T}, T\} + \max\{\frac{\gamma^{\hat{T}-T}-1}{1-\gamma}, 0\} \right))
\end{aligned}
$$

where $\alpha' = 2\eta_0\alpha$.

First, we consider $T \le \hat{T}$. Use $T = \frac{1}{\ln(1/\gamma)} \ln\left(1 + \frac{\eta_0}{\kappa\alpha}\right) = \left\lceil \mathcal{O}\left(\frac{\kappa}{\eta_0} \ln\left(1 + \frac{\eta_0}{\kappa\alpha}\right)\right) \right\rceil$ to get

$$
\begin{aligned}
\text{ERUB} &\le \left( \frac{\alpha}{\alpha + \eta_0/\kappa} \right) \left( 1 + \alpha'\gamma^{-1}(\frac{2}{\ln(1/\gamma)} \ln\left(1 + \frac{\eta_0}{\kappa\alpha}\right))^2 \right) \\
&\le \left( \frac{\alpha}{\alpha + \eta_0/\kappa} \right) \left( 1 + \frac{8\kappa^2\alpha}{\eta_0\gamma} \ln^2\left(1 + \frac{\eta_0}{\kappa\alpha}\right) \right) \\
&\le \mathcal{O}\left( \frac{\kappa}{\kappa + \eta_0/\alpha} \left( 1 + \frac{8\kappa^2\alpha}{\eta_0\gamma} \ln^2\left(1 + \frac{\eta_0}{\kappa\alpha}\right) \right) \right) \\
&= \mathcal{O}\left( \frac{\kappa}{\kappa + \eta_0/\alpha} \left( 1 + \frac{4\kappa}{\gamma} \right) \right) \\
&= \mathcal{O}\left( \frac{\kappa^2}{\kappa + \eta_0/\alpha} \right)
\end{aligned}
$$

where we used $\ln(1/\gamma) \geq \eta_0/\kappa$ and $\ln(1 + x) \leq \sqrt{x}$ for any $x > 0$.

Second, when $T > \hat{T}$,

$$\text{ERUB} \leq \gamma^T(1 + \frac{\alpha'}{\gamma} T \left( \hat{T} + \frac{\gamma^{\hat{T}-T} - 1}{1 - \gamma} \right))$$

$$\leq \mathcal{O} \left( \gamma^T + \frac{2\alpha'}{\gamma} T \kappa (\gamma^{\hat{T}} - \gamma^T) \right).$$

Make use of $T = \left\lceil \frac{1}{\ln(1/\gamma)} \ln \left( 1 + \frac{\eta_0}{\kappa\alpha} \right) \right\rceil$ to show

$$\text{ERUB} \leq \mathcal{O} \left( \frac{\kappa}{\kappa + \eta_0/\alpha} + \frac{4\kappa^2\alpha}{\eta_0\gamma}(\gamma^{\hat{T}} - \frac{\kappa}{\kappa + \eta_0/\alpha}) \ln \left( 1 + \frac{\eta_0}{\kappa\alpha} \right) \right)$$

$$\leq \mathcal{O} \left( \frac{\kappa^2}{\kappa + \eta_0/\alpha} \gamma^{\hat{T}-1} \ln \left( 1 + \frac{\eta_0}{\kappa\alpha} \right) \right).$$

$\square$

*Proof of Theorem 4.6.* By Lemma B.7, we can rewrite $U_3$ as

$$U_3 = \sum_{t=1}^{T} \gamma^{T-t} \frac{(1-\beta)^2}{(1-\beta^t)^2} \sum_{i=1}^{t} \beta^{2(t-i)} \sigma_i^2$$

$$\leq \sum_{t=1}^{\hat{T}} \gamma^{T-t} \gamma^{2(t-1)} \sum_{i=1}^{t} \beta^{2(t-i)} \sigma_i^2 + \sum_{t=\hat{T}+1}^{T} \gamma^{T-t} \gamma^{2(\hat{T}-1)} \sum_{i=1}^{t} \beta^{2(t-i)} \sigma_i^2$$

$$\leq \gamma^{T-\hat{T}} \underbrace{\sum_{t=1}^{\hat{T}} \gamma^{\hat{T}-t} \gamma^{2(t-1)} \sum_{i=1}^{t} \beta^{2(t-i)} \sigma_i^2}_{V_1} + \gamma^{2(\hat{T}-1)} \underbrace{\sum_{t=\hat{T}+1}^{T} \gamma^{T-t} \sum_{i=1}^{t} \beta^{2(t-i)} \sigma_i^2}_{V_2}$$

We derive $V_1$ and $V_2$ separately.

For $V_1$, we can obtain the upper bound by

$$V_1 = \sum_{t=1}^{\hat{T}} \gamma^{\hat{T}-t} \gamma^{2(t-1)} \sum_{i=1}^{t} \beta^{2(t-i)} \sigma_i^2$$

$$= \gamma^{\hat{T}-2} \sum_{t=1}^{\hat{T}} \gamma^{t} \sum_{i=1}^{t} \beta^{2(t-i)} \sigma_i^2$$

$$= \gamma^{\hat{T}-2} \sum_{i=1}^{\hat{T}} \beta^{-2i} \sigma_i^2 \sum_{t=i}^{\hat{T}} \left( \gamma\beta^2 \right)^t$$

$$= \gamma^{\hat{T}-2} \sum_{i=1}^{\hat{T}} \beta^{-2i} \sigma_i^2 \frac{\left( \gamma\beta^2 \right)^i - \left( \gamma\beta^2 \right)^{\hat{T}+1}}{1 - \gamma\beta^2}$$

$$= \gamma^{2\hat{T}-3} \sum_{i=1}^{\hat{T}} \frac{\gamma^{i-\hat{T}-1} - \beta^{2(\hat{T}+1-i)}}{1 - \gamma\beta^2} \sigma_i^2$$

$$= \gamma^{2\hat{T}-3} \sum_{i=1}^{\hat{T}} \frac{1 - (\gamma\beta^2)^{\hat{T}+1-i}}{1 - \gamma\beta^2} \gamma^{i-\hat{T}-1} \sigma_i^2$$

$$\leq \frac{\gamma^{\hat{T}}}{\gamma^2(1 - \gamma\beta^2)} \sum_{i=1}^{\hat{T}} \gamma^{i} \sigma_i^2$$

$$\leq \frac{\gamma^{\hat{T}}}{\gamma(\gamma - \beta^2)} \sum_{i=1}^{\hat{T}} \gamma^{i} \sigma_i^2$$

For $V_2$, we can derive

$$V_2 = \sum_{t=\hat{T}+1}^{T} \gamma^{T-t} \sum_{i=1}^{t} \beta^{2(t-i)} \sigma_i^2$$

$$= \sum_{t=1}^{T} \gamma^{T-t} \sum_{i=1}^{t} \beta^{2(t-i)} \sigma_i^2 - \sum_{t=1}^{\hat{T}} \gamma^{T-t} \sum_{i=1}^{t} \beta^{2(t-i)} \sigma_i^2$$

$$= \sum_{t=1}^{T} \gamma^{T-t} \sum_{i=1}^{t} \beta^{2(t-i)} \sigma_i^2 - \gamma^{T-\hat{T}} \sum_{t=1}^{\hat{T}} \gamma^{\hat{T}-t} \sum_{i=1}^{t} \beta^{2(t-i)} \sigma_i^2.$$

We first consider the first term

$$\sum_{t=1}^{T} \gamma^{T-t} \sum_{i=1}^{t} \beta^{2(t-i)} \sigma_i^2$$

$$= \sum_{i=1}^{T} \sigma_i^2 \sum_{t=i}^{T} \gamma^{T-t} \beta^{2(t-i)}$$

$$= \sum_{i=1}^{T} \gamma^T \beta^{-2i} \sigma_i^2 \sum_{t=i}^{T} \gamma^{-t} \beta^{2t}$$

$$= \sum_{i=1}^{T} \gamma^T \beta^{-2i} \sigma_i^2 \frac{(\beta^2/\gamma)^i - (\beta^2/\gamma)^{T+1}}{1 - (\beta^2/\gamma)}$$

$$= \sum_{i=1}^{T} \frac{\gamma^{T+1-i} - \beta^{2(T+1-i)}}{\gamma - \beta^2} \sigma_i^2.$$

Similarly, we have

$$\gamma^{T-\hat{T}} \sum_{t=1}^{\hat{T}} \gamma^{\hat{T}-t} \sum_{i=1}^{t} \beta^{2(t-i)} \sigma_i^2$$

$$= \gamma^{T-\hat{T}} \sum_{i=1}^{\hat{T}} \frac{\gamma^{\hat{T}+1-i} - \beta^{2(\hat{T}+1-i)}}{\gamma - \beta^2} \sigma_i^2$$

$$= \sum_{i=1}^{\hat{T}} \frac{\gamma^{T+1-i} - \gamma^{T-\hat{T}} \beta^{2(\hat{T}+1-i)}}{\gamma - \beta^2} \sigma_i^2.$$

Thus,

$$V_2 = \sum_{i=1}^{T} \frac{\gamma^{T+1-i} - \beta^{2(T+1-i)}}{\gamma - \beta^2} \sigma_i^2 - \sum_{i=1}^{\hat{T}} \frac{\gamma^{T+1-i} - \gamma^{T-\hat{T}} \beta^{2(\hat{T}+1-i)}}{\gamma - \beta^2} \sigma_i^2$$

$$= \sum_{i=\hat{T}+1}^{T} \frac{\gamma^{T+1-i} - \beta^{2(T+1-i)}}{\gamma - \beta^2} \sigma_i^2 + \sum_{i=1}^{\hat{T}} \frac{\gamma^{T-\hat{T}} - \beta^{2(T-\hat{T})}}{\gamma - \beta^2} \beta^{2(\hat{T}+1-i)} \sigma_i^2$$

$$\leq \sum_{i=\hat{T}+1}^{T} \frac{\gamma^{T+1-i} - \beta^{2(T+1-i)}}{\gamma - \beta^2} \sigma_i^2 + \sum_{i=1}^{\hat{T}} \frac{\gamma^{T-\hat{T}}}{\gamma - \beta^2} \beta^{2(\hat{T}+1-i)} \sigma_i^2.$$

Substitute $V_1$ and $V_2$ into $U_3$ to get

$$U_3 \leq \gamma^T \frac{1}{\gamma(\gamma - \beta^2)} \sum_{i=1}^{\hat{T}} \gamma^i \sigma_i^2 + \gamma^{2\hat{T}-2} \sum_{i=\hat{T}+1}^{T} \frac{\gamma^{T+1-i} - \beta^{2(T+1-i)}}{\gamma - \beta^2} \sigma_i^2$$

$$+ \sum_{i=1}^{\hat{T}} \frac{\gamma^{T+\hat{T}-2}}{\gamma - \beta^2} \beta^{2(\hat{T}+1-i)} \sigma_i^2$$

$$\leq \left( \frac{\gamma^T}{\gamma(\gamma - \beta^2)} \sum_{i=1}^{\hat{T}} (\gamma^i + \gamma^{\hat{T}-1} \beta^{2(\hat{T}+1-i)}) \sigma_i^2 + \gamma^{2\hat{T}-2} \sum_{i=\hat{T}+1}^{T} \frac{\gamma^{T+1-i} - \beta^{2(T+1-i)}}{\gamma - \beta^2} \sigma_i^2 \right)$$

$$\leq \left( \frac{2\gamma^T}{\gamma(\gamma - \beta^2)} \sum_{i=1}^{\hat{T}} \gamma^i \sigma_i^2 + \gamma^{2\hat{T}-2} \sum_{i=\hat{T}+1}^{T} \frac{\gamma^{T+1-i} - \beta^{2(T+1-i)}}{\gamma - \beta^2} \sigma_i^2 \right)$$

$$= \sum_{t=1}^{T} q_t \sigma_t^2$$

where

$$q_t = \frac{2}{\gamma(\gamma - \beta^2)} \gamma^{T+t} \mathbb{I}_{T \leq \hat{T}} + \gamma^{2(\hat{T}-1)} \frac{\gamma^{T+1-i} - \beta^{2(T+1-i)}}{\gamma - \beta^2} \gamma^{T-t} \mathbb{I}_{T > \hat{T}}$$

$$\leq c_1 \gamma^{T+t} \mathbb{I}_{T \leq \hat{T}} + \gamma^{\hat{T}-1} c_2 \gamma^{T-t} \mathbb{I}_{T > \hat{T}}$$

where $c_1 = \frac{2}{\gamma(\gamma - \beta^2)}$ and $c_2 = \frac{\gamma^{2\hat{T}}}{\gamma - \beta^2}$.

When $T > \hat{T}$, by Lemma 4.1, the lower bound of $R\sum_{t=1}^{T} q_t \sigma_t^2$ is

$$\left(\sum_{t=1}^{T} \sqrt{q_t}\right)^2 = \gamma^T \left(\sum_{t=1}^{\hat{T}} \sqrt{c_1 \gamma^t} + \sum_{t=\hat{T}+1}^{T} \sqrt{\gamma^{\hat{T}-1} c_2 \gamma^{-t}}\right)^2$$

$$= \gamma^T \left(\sqrt{c_1 \gamma} \frac{1 - \gamma^{\hat{T}/2}}{1 - \sqrt{\gamma}} + \sqrt{c_2} \frac{1 - \gamma^{(\hat{T}-T-1)/2}}{\sqrt{\gamma} - 1}\right)^2$$

$$= \gamma^T \left(\sqrt{c_1 \gamma} \frac{1 - \gamma^{\hat{T}/2}}{1 - \sqrt{\gamma}} + \sqrt{c_2} \frac{\gamma^{(\hat{T}-T-1)/2} - 1}{1 - \sqrt{\gamma}}\right)^2$$

$$\leq \mathcal{O}\left(c_2 \left\{\frac{\gamma^{(\hat{T}-1)/2} - \gamma^{T/2}}{1 - \sqrt{\gamma}}\right\}^2\right)$$

which is achieved when

$$\sigma_t^2 = \frac{1}{R} \sum_{i=1}^{T} \sqrt{\frac{q_i}{q_t}}.$$

By Theorem 4.4, because $\zeta \geq 0$, we have

$$\text{ERUB} \leq \gamma^T + 2R\eta_0 \alpha U_3$$

$$= \gamma^T + 2\eta_0 \alpha \sum_{t=1}^{T} Rq_t \sigma_t^2.$$

And the minimum of the upper bound is

$$\text{ERUB}_{\min} = \gamma^T + \alpha' \mathcal{O}\left(\left\{\frac{\gamma^{(\hat{T}-1)/2} - \gamma^{T/2}}{1 - \sqrt{\gamma}}\right\}^2\right)$$

where $\alpha' = 2\eta_0 c_2 \alpha$. Let $T = \frac{2}{\ln(1/\gamma)} \ln\left(1 + \frac{\eta_0}{\kappa \alpha}\right)$. Then,

$$\text{ERUB}_{\min} = \mathcal{O}\left(\left(\frac{\kappa\alpha}{\kappa\alpha + \eta_0}\right)^2 + \frac{\alpha'}{(1 - \sqrt{\gamma})^2}\left\{\frac{\gamma^{(\hat{T}-1)/2} - (1 - \gamma^{(\hat{T}-1)/2})\kappa\alpha}{\kappa\alpha + \eta_0}\right\}^2\right)$$

$$\leq \mathcal{O}\left(\left(\frac{\kappa\alpha}{\kappa\alpha + \eta_0}\right)^2 + \frac{2\eta_0 c_2 \alpha}{(1 - \sqrt{\gamma})^2}\left\{\frac{\gamma^{(\hat{T}-1)/2}}{\kappa\alpha + \eta_0}\right\}^2\right)$$

$$= \mathcal{O}\left(\frac{\kappa\alpha}{(\kappa\alpha + \eta_0)^2}\left(\kappa\alpha + \frac{2\eta_0 c_2/\kappa}{(1 - \sqrt{\gamma})^2} \gamma^{(\hat{T}-1)}\right)\right)$$

$$= \mathcal{O}\left(\frac{\kappa\alpha}{(\kappa\alpha + \eta_0)^2}\left(\kappa\alpha + c_3 \eta_0\right)\right)$$

$$\leq \mathcal{O}\left(\frac{\kappa\alpha}{\kappa\alpha + \eta_0}\right)$$

where $c_3$ is some constant.

When $T \leq \hat{T}$,

$$U_3 \leq \gamma^{T-T} \underbrace{\sum_{t=1}^{T} \gamma^{T-t} \gamma^{2(t-1)} \sum_{i=1}^{t} \beta^{2(t-i)} \sigma_i^2}_{V_1}$$

$$\leq \frac{\gamma^{T-2}}{1 - \gamma\beta^2} \sum_{i=1}^{T} \gamma^i \sigma_i^2$$

with which we obtain

$$\text{ERUB} \leq \gamma^T + 2R\eta_0 \alpha U_3$$

$$\leq \gamma^T + 2\eta_0 \alpha \frac{\gamma^{-2}}{1 - \gamma\beta^2} \sum_{t=1}^{T} Rq_t \sigma_t^2.$$

where we let $q_t = \gamma^{T+t}$. By Lemma 4.1,

$$\sum_{i=1}^{T} R q_t \sigma_i^2 \geq \left( \sum_{t=1}^{T} \sqrt{q_t} \right)^2$$

$$= \gamma^T \left( \sum_{t=1}^{T} \gamma^{t/2} \right)^2$$

$$= \gamma^{T+1} \left( \frac{1 - \gamma^{T/2}}{1 - \sqrt{\gamma}} \right)^2.$$

Thus,

$$\text{ERUB}_{\min} \leq \gamma^T + 2\eta_0 \alpha \frac{\gamma^{T-1}}{1 - \gamma\beta^2} \left( \frac{1 - \gamma^{T/2}}{1 - \sqrt{\gamma}} \right)^2$$

$$= \gamma^T \left( 1 + 2\eta_0 \gamma c_1 \alpha \left( \frac{1 - \gamma^{T/2}}{1 - \sqrt{\gamma}} \right)^2 \right)$$

Let $T = \left\lceil \frac{2}{\ln(1/\gamma)} \ln\left(1 + \frac{\eta_0}{\kappa\alpha}\right) \right\rceil$. Then,

$$\text{ERUB}_{\min} \leq \left( \frac{\kappa\alpha}{\kappa\alpha + \eta_0} \right)^2 \left( 1 + \frac{2\eta_0 \gamma c_1 \alpha}{(1 - \sqrt{\gamma})^2} \left( \frac{1}{\kappa\alpha + 1} \right)^2 \right)$$

$$\leq \left( \frac{\kappa\alpha}{\kappa\alpha + \eta_0} \right)^2 \left( 1 + \mathcal{O}\left( \frac{1}{\kappa\alpha + 1} \right) \right)$$

$$\leq \mathcal{O}\left( \frac{\kappa\alpha}{\kappa\alpha + \eta_0} \right)^2.$$

In summary,

$$\text{ERUB}_{\min} \leq \mathcal{O}\left( \frac{\kappa\alpha}{\kappa\alpha + \eta_0} \left( \mathbb{I}_{T \leq \hat{T}} \frac{\kappa\alpha}{\kappa\alpha + \eta_0} + \mathbb{I}_{T > \hat{T}} \right) \right)$$

$\square$

## C.3 STOCHASTIC GRADIENT DESCENTS

*Proof of Theorem 4.7.* Let $\tilde{\nabla}_t$ be the stochastic gradient of the step $t$. By the smoothness, we have

$$f(\theta_{t+1}) - f(\theta_t) \leq -\eta_t \nabla_t^\top (\tilde{\nabla}_t + G\sigma_t \nu_t/n) + \frac{1}{2} M \eta_t^2 \left\| \tilde{\nabla}_t + G\sigma_t \nu_t/n \right\|^2$$

$$= -\eta_t \nabla_t^\top (\nabla_t + \sigma_g \xi_t/n + G\sigma_t \nu_t/n) + \frac{1}{2} M \eta_t^2 \left\| \nabla_t + \sigma_g \xi_t/n + G\sigma_t \nu_t/n \right\|^2.$$

Note that $\mathbb{E}(\sigma_g \xi_t/n + G\sigma_t \nu_t/n) = 0$ and $\mathbb{E}(\sigma_g \xi_t/n + G\sigma_t \nu_t/n)^2 = \sigma_g^2 + (G\sigma_t/n)^2$. Without loss of generality, we can write $\sigma_g \xi_t + G\sigma_t \nu_t$ as $\tilde{\sigma}_t \zeta_t$ where $\tilde{\sigma}_t \triangleq \sqrt{\sigma_g^2 + (G\sigma_t)^2}$ and $\zeta_t$ is a random vector with $\mathbb{E}\zeta_t = 0$ and $\mathbb{E}\left\| \zeta_t \right\|^2 \leq D$. Therefore,

$$f(\theta_{t+1}) - f(\theta_t) \leq -\eta_t \nabla_t^\top (\nabla_t + \tilde{\sigma}_t \zeta_t/n) + \frac{1}{2} M \eta_t^2 \left\| \nabla_t + \tilde{\sigma}_t \zeta_t/n \right\|^2$$

$$= -\eta_t (1 - \frac{1}{2} M \eta_t) \left\| \nabla_t \right\|^2 - (1 - M\eta_t) \eta_t \nabla_t^\top \tilde{\sigma}_t \zeta_t/n + \frac{1}{2} M \eta_t^2 \left\| \tilde{\sigma}_t \zeta_t/n \right\|^2$$

$$\leq -2\mu\eta_t (1 - \frac{1}{2} M \eta_t)(f(\theta_t) - f(\theta^*)) - (1 - M\eta_t) \eta_t \nabla_t^\top \tilde{\sigma}_t \zeta_t/n$$

$$+ \frac{1}{2} M \eta_t^2 \left\| \tilde{\sigma}_t \zeta_t/n \right\|^2.$$

Then following the same proof of Theorem 4.1, we can get

$$
\mathbb{E}[f(\theta_{T+1})] - f(\theta^*) \leq \gamma^T(\mathbb{E}[f(\theta_1)] - f(\theta^*)) + R' \sum_{t=1}^{T} \gamma^{T-t} \alpha \frac{1}{G^2} \tilde{\sigma}_t^2 (\mathbb{E}[f(\theta_1)] - f(\theta^*))
$$

$$
= \left[ \gamma^T + R' \sum_{t=1}^{T} \gamma^{T-t} \alpha (\frac{1}{G^2}\sigma_g^2 + \sigma_t^2) \right] (\mathbb{E}[f(\theta_1)] - f(\theta^*))
$$

$$
= \left[ \gamma^T + R'\alpha \frac{1}{G^2}\sigma_g^2 \frac{1-\gamma^T}{1-\gamma} + R' \sum_{t=1}^{T} \gamma^{T-t}\alpha\sigma_t^2 \right] (\mathbb{E}[f(\theta_1)] - f(\theta^*))
$$

$$
\leq \left[ \gamma^T + \frac{R'\kappa\alpha}{G^2}\sigma_g^2 + R' \sum_{t=1}^{T} \gamma^{T-t}\alpha\sigma_t^2 \right] (\mathbb{E}[f(\theta_1)] - f(\theta^*)).
$$

where $\frac{R'\kappa\alpha}{G^2} = \frac{D}{2\mu(f(\theta_1)-f(\theta^*))}\frac{1}{n^2} = \frac{D}{2\mu(f(\theta_1)-f(\theta^*))}\min\{\frac{1}{N^2R},1\} \leq \frac{D}{2\mu(f(\theta_1)-f(\theta^*))}\frac{1}{N^2R}$. $\qquad\square$

*Proof of Theorem 4.8.* Without loss of generality, we can write $\sigma_g\xi_t + G\sigma_t\nu_t$ as $\tilde{\sigma}_t\zeta_t$ where $\tilde{\sigma}_t \triangleq \sqrt{\sigma_g^2 + (G\sigma_t)^2}$ and $\zeta_t$ is a random vector with $\mathbb{E}\zeta_t = 0$ and $\mathbb{E}\|\zeta_t\|^2 \leq D$. Therefore, we replace $\nu_t$ by $\zeta_t$ and $\sigma_t^2$ by $\tilde{\sigma}_t^2/G^2 = \sigma_g^2/G^2 + \sigma_t^2$. Now, we only need to update $U_3(\sigma, T)$ as

$$
\tilde{U}_3 = \frac{1}{G^2} \sum_{t=1}^{T} \gamma^{T-t}\frac{(1-\beta)^2}{(1-\beta^t)^2} \sum_{i=1}^{t} \beta^{2(t-i)}\tilde{\sigma}_i^2
$$

$$
= \sum_{t=1}^{T} \gamma^{T-t}\frac{(1-\beta)^2}{(1-\beta^t)^2} \sum_{i=1}^{t} \beta^{2(t-i)}(\frac{1}{G^2}\sigma_g^2 + \sigma_t^2)
$$

$$
= U_3^g + U_3
$$

where we define

$$
U_3^g \triangleq \frac{1}{G^2}\sigma_g^2 \sum_{t=1}^{T} \gamma^{T-t}\frac{(1-\beta)^2}{(1-\beta^t)^2} \sum_{i=1}^{t} \beta^{2(t-i)}.
$$

We can upper bound $U_3^g$ by

$$
U_3^g = \frac{1}{G^2}\sigma_g^2 \sum_{t=1}^{T} \gamma^{T-t}\frac{(1-\beta)^2}{(1-\beta^t)^2}\frac{1-\beta^{2t}}{1-\beta^2}
$$

$$
= \frac{1}{G^2}\sigma_g^2 \sum_{t=1}^{T} \gamma^{T-t}\frac{1-\beta}{1-\beta^t}\frac{1+\beta^t}{1+\beta}
$$

$$
\leq \frac{1}{G^2}\sigma_g^2 \sum_{t=1}^{T} \gamma^{T-t}
$$

$$
\leq \frac{1}{G^2}\sigma_g^2 \frac{1}{1-\gamma}
$$

$$
= \frac{1}{G^2}\kappa\sigma_g^2.
$$

Combine with the factors of $U_3$ in the PGD bounds:

$$
\alpha R' U_3^g \leq \frac{\alpha R'}{G^2}\kappa\sigma_g^2 = \frac{\alpha R'}{G^2}\kappa\sigma_g^2 = \frac{D\sigma_g^2}{2\mu n^2(f(\theta_1) - f(\theta^*))} \leq \frac{D\sigma_g^2}{2\mu N^2 R(f(\theta_1) - f(\theta^*))}.
$$

$\qquad\square$

# D EXPERIMENTS

**Dataset**. **(1) Synthetic data**. We generate a 100-dimensional dataset including linearly separable data points using `sckit-learn` package. The data points are distributed in two points in the hyper-cubic with Euclidean distance of 10. In total, 1000 samples are generated for training with the

logistic loss. **(2) Real data**. We create a subset of the MNIST dataset (Lecun et al., 1998) including 1000 handwritten images of digit 3 and 5 (MNIST35). Compared to the original dataset ($70,000$ samples), the small set will be more vulnerable to attack and the private learning will require larger noise (see the $1/N$ factor in Eq. (1)). Following the preprocessing in (Abadi et al., 2016), we project the vectorized images into a 60-dimensional subspace extracted by PCA.

**Setup**. The samples are first normalized so that $\sum_{n=1}^{N} x_n = 0$ and the standard deviation is 1. Then the sample norms are scaled such that $\max_n \|x_n\| = 10$ (i.e., *data scales*). We fix the learning rate to $0.1$ based on the corresponding experiments of non-private training (same setting without noise). The total privacy budget is $0.1963$-zCDP (equal to $(4, 10^{-8})$-DP) which implies $R = 0.3927$. To control the sensitivity of the gradients, we clip gradients by a clipping norm fixed at 4. Formally, we scale down the sample gradients to length 4 if its norm is larger than 4. Because the schedule highly depends on the iteration number $T$, we grid search the best $T$ for compared methods. Therefore, we ignore the privacy cost of such tuning in our experiments which protocol is also used in previous work (Abadi et al., 2016; Wu et al., 2017). All the experiments are repeated 100 times and metrics are averaged afterwards.

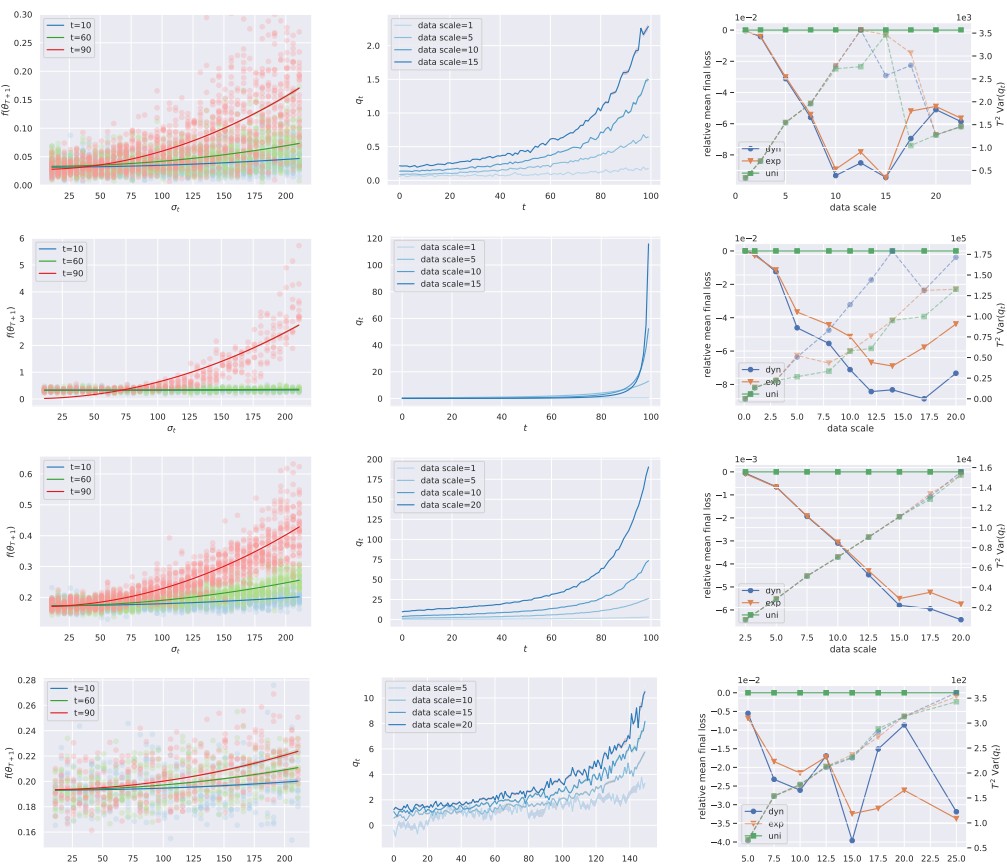

Figure 3: Experiments with Logistic loss on synthetic data and squared, Logistic loss and DNN on MNIST35 by rows. (**left**) The final loss is fitted by a quadratic function formulated as $c_2 \sigma_t^2 + c_0$. (**middle**) The influence values are estimated using the Hessian eigenvalues (squared loss) and by retraining (logistic loss). The larger is the data scale, the larger the influence variance. (**right**) The relative final training losses versus the data scale where uniform schedule (uni), dynamic schedule (dyn) and exponential-decaying schedule (exp) are compared. The relative loss is computed w.r.t. the losses of the uniform schedule. For example, if the dynamic loss is $e_1$ and the uniform loss is $e_0$, then the relative loss is $(e_1 - e_0)/e_0$. The dashed lines are with the right axis to present influence-related term.

**Estimate of influences by retraining**. In the left pane of Fig. 3, we estimate the influence of $\sigma_t$ by retraining multiple times. When keeping $\sigma_i$ for $i \neq t$ fixed (as the fine-tuend uniform schedule $\sigma$), the value of $\sigma_t$ is varied from 20 to 200. Then we fit a quadratic function of $\sigma_t$ where the coefficient is treated as estimation of $q_t$. Repeating the estimation for all $t$ in range 1 to 100 could provide us the trend of influence in the middle pane.

**Results**. In the left pane of Fig. 3, the squared final loss (i.e., $f(\theta_{T+1})$) is approximately a quadratic function of the $\sigma_t$ with small relative variance (i.e., the variance divided by the mean value, $0.14, 0.032, 0.016$ for $t = 10, 60, 90$, respectively). When $\sigma_t$ increases, the frequency of clipping increases as well which leads to more variance in the final losses. We use the least square method to fit the relation shown as the solid lines. The final logistic loss is more sensitive to the noise because of the additional uncertainty from the changing Hessian. We still fit a quadratic function on $\sigma_t$. It turns out the relative variances of the quadratic coefficient (approximately the influence) is small which are $0.025, 0.024, 0.027$ for $t = 10, 60, 90$, respectively.

The middle pane shows the relationship between the estimated $q_t$ and learning steps. The $q_t$ of squared loss is computed by analytical solution using the Hessian eigen values. The $q_t$ of logistic loss is computed by retraining based on uniform schedule. Both loss functions show an increased influence as learning continues, which indicates that the dynamic schedule should be decreasing accordingly. The squared loss has a rather steep trend while the logistic has a relative flatten one. The reason is that the logistic loss has a larger variance in the gradient norm and therefore clipping happens more frequently (approximately more than $80\%$ gradients are clipped). As a result, the variance of influence will be relative small for logistic losses. Moreover, we vary the data scale (scale all samples uniformly such that all sample norms are less than a specific value), changing the variance of influence, as seen in the figures.

The last pane compares uniform, dynamic and exponential decay schedules Yu et al. (2019) using final losses relative to the uniform schedule. We set the exponential decay schedule to approximate the dynamic schedule by fitting $\hat{\sigma}_t = \sigma_0 \exp(-kt)$ using the least squares method. The final losses are picked by grid searching the best $T \in [1, 100]$. We see that the advantage of the dynamic schedule over the uniform one increases when data scales less than 15. But we also notice some loss gaps decrease, suggesting that the data scale is not the inherent reason for the dynamic advantage. According to Lemma 4.1, the advantage should be proportional to $T^2 \text{Var}(q_t)$ which is shown to be decreasing when the data scale is larger than 15. When the data scale continues to increase, gradient clipping will change the curvature of the loss function. Therefore, a increasing $\text{Var}(Tq_t)$ is witnessed. Meanwhile, the loss gap decreases.

In the last row of Fig. 3, the DNN is experimented with the same setting. Though the influence increases by $t$, the variance is small and less dependent on the data scale in comparison to shallow models. Though the dynamic schedule estimated by retraining influences does not performs stably, the *exp* method still decreasingly depends on the variance of influence as expectation.

