# OpenReview forum: "On Dynamic Noise Influence in Differential Private Learning"
_ICLR.cc/2021/Conference — Reject_

### Official Review · AnonReviewer4 · 2020-10-26
**Important Theoretical Contribution with Practical Relevance, Exposition Needs Work.**

**Rating:** 6
**Confidence:** 4

**Review:**

Summary


Gradient Descent and related variants are the defacto standard algorithms for optimizing empirical risk functions. Since published models have been shown in the literature to leak private information, the problem of performing gradient descent under privacy constraints is an important one. Given a fixed privacy budget R, private gradient descent adds noise to the gradients at each round, ensuring overall privacy budget R by composition. However, this still leaves the question of what privacy schedule is best open, since any schedule whose privacy budgets sum up to R achieves the privacy objective. In this paper they compute the optimal privacy schedule from an accuracy perspective via a novel analysis of the convergence of private gradient descent on loss functions that satisfy the PL condition, which turns out to be exponentially decaying noise (increasing privacy budget for each round). These results extend to a privatized variant of the momentum-based gradient descent algorithm, although the dynamic privacy schedule has less improvement there. Experimental results show that dynamic privacy schedules lead to enhanced accuracy even absent convexity.

Pros
- improves over previous soa analysis of private gradient descent (wang 2017) by $log N^2$ factor
- Novel analysis for an important practical problem that also yields intuition via the notion of noise influence / propagation

Cons
- Grammatical Errors throughout: e.g. sentence fragment page 6 “because our bound saved a factor of log N and is thus tighter, as compared to Wang”
- Overall the clarity of the writing and explanations, both in terms of prefacing the technical content, and readability could be improved.
- Experimental section needs more detailed fleshing out: 1) how does the middle pane show the trend of variance of influence? 2) how are we modeling influence for DNN and how are we then computing the optimal dynamic schedule 3) does plot #1 perform a sensitivity analysis on the ERR via retraining to estimate $q_t$ empirically? Explain this.


Comments
- why does the utility bound for non-private GD in table 1 have a factor of $R$ (privacy budget) in the denominator?
- What is the difference between $v_t$ and $g_t$? $v_t = g_t$ it says on the bottom of page 3
- $v_t$ and $\nu_t$ look very similar, choose more distinct notation
- write out the definition of the ERUB in the main text on page 4
- 4.11 uses the improved analysis to sharpen the utility analysis of a uniform schedule by choosing an optimal value of T. Please contrast the Wang bound more clearly with (5) and state the improvement factor. We can read this from the analysis but it should be stated more prominently.
- Discussion of robustness on page 6 is a little imprecisely worded. Is the takeaway that when the condition number is very high relative to the sample size, the dynamic schedule ERUB doesn’t blow up? Since we are calling this robustness, is it obvious that noise in the samples or labels leads to high loss curvature?

---

> ### Author Response · Authors · 2020-11-16
> **Response to Reviewer #4**
>
> **Q1**: _Grammatical Errors throughout: e.g. sentence fragment page 6 “because our bound saved a factor of log N and is thus tighter, as compared to Wang”_
>
> **A1**: Thanks for the detailed comment. We fix the language problem and proofread our paper again.
>
> **Q2**: _How does the middle pane show the trend of variance of influence?_
>
> **A2**: The dashed lines (coordinated by the right axis) represents the $T^2$ times the variance of influences $q_t$ through iterations. Formally, we compute the variance by $\sum_{t=1}^T (q_t - \bar q_t)^2 / T$. To clarify, we explain the variance of influence in Eq. (22) and the experiment part (highlighted).
>
> **Q3**: _how are we modeling influence for DNN and how are we then computing the optimal dynamic schedule_
>
> **A3**: For DNN, we retrain the models to estimate the influence of noise on the final loss. Details of the estimation method are in Appendix D (see highlights by blue). When the influence trend is estimated, we compute the dynamic schedule by Lemma 4.1. Note the schedule is only optimal under the PL condition. But here we empirically check if the results hold when the PL condition is not exactly satisfied.
>
> **Q4**: _does plot #1 perform a sensitivity analysis on the ERR via retraining to estimate $q_t$ empirically? Explain this._
>
> **A4**: Yes. You may think of the influence estimation as the sensitivity analysis for each $\sigma_t$. The $q_t$ is estimated by the quadratic coefficient of the final loss on the $\sigma_t$. In our revision, we elaborate on the estimation method at the end of page 28 (texts highlighted by blue color).
>
> **Q5**: _why does the utility bound for non-private GD in table 1 have a factor of  R  (privacy budget) in the denominator?_
>
> **A5**: Generally for non-private gradient descent, the loss will converge to $0$ when $T$ is infinitely large. In comparison, private algorithms which cannot run that much iterations are not comparable. To conduct a fair comparison of the bound, we limit the total number of iterations of all algorithms, $T$, to be $O(\ln (D/(N^2 R)))$ which is also used by the private gradient descent algorithms. Therefore, the final bound includes the $R$ term.
>
> **Q6**: _What is the difference between $v_t$ and  g_t?? it says on the bottom of page 3 $v_t$ and $\nu_t$ look very similar, choose more distinct notation. write out the definition of the ERUB in the main text on page 4_
>
> **A6**: Sorry for the confusion. The $g_t$ represents the noised gradients in Line 4 of Alg. 1. Instead, $v_t=(m_t, g_t)$ generally represents the updates of model. For example, the momentum method has $v_t$ different from $g_t$ at the bottom of Page 3. To avoid confusion, we update $v_t$ by $\phi_t$ at the end of page 3 (highlighted by blue color).
>
> **Q7**: _4.1.1 uses the improved analysis to sharpen the utility analysis of a uniform schedule by choosing an optimal value of T. Please contrast the Wang bound more clearly with (5) and state the improvement factor. We can read this from the analysis but it should be stated more prominently._
>
> **A7**: The comparison is briefly done in Table 1 where $\ln^2 N$ (Wang et al., 2017) is replaced by $\ln N$. The difference is because they relax the $\sum_{t=1}^T \gamma^{T-t}$ to $T$ while we use a tighter bound $1 - \gamma^T$. Thanks for the suggestion, we provide a detailed explanation in the revision. Also, in the sketch of the proof of Theorem 4.2, we discussed why (Wang et al., 2017)'s bound is less tight.
>
>
> **Q8**: _Discussion of robustness on page 6 is a little imprecisely worded. Is the takeaway that when the condition number is very high relative to the sample size, the dynamic schedule ERUB doesn’t blow up? Since we are calling this robustness, is it obvious that noise in the samples or labels leads to high loss curvature?_
>
> **A8**: Thanks for pointing out this. Initially, we want to express the instability of the gradient descent when the condition number is large. For private algorithms, the learning rate cannot be extremely small, otherwise, the learning cannot make any progress when the $T$ is finite. Therefore, a large condition number will result in significant fluctuation of the noised gradient descent with a non-infinitesimal learning rate. That is why we call the issue a robustness.
>
> To clarify the concept, we update at the end of page 6 which are highlighted. In addition, we add a figure for the illustration of the issue.

---

### Official Review · AnonReviewer2 · 2020-10-28
**Paper studies use of momentum based method to improve utility of private-SGD**

**Rating:** 4
**Confidence:** 3

**Review:**

The paper studies private gradient descent when the noise added to each of the iteration is dynamically scheduled. Prior to this work, the work of Zhou et al. tries to achieve the same for DP-SGD and they analyze their algorithm for many variants of adaptive gradient descent based method. The difference with Zhou et al. is that they do not gradient norm and the generalization property of DP. As a result, the authors claim that Zhou et al. achieves suboptimal utility guarantee.

I will keep my reviews limited to the current submission. My first objection to the paper is that gradient descent requires computation of gradients over the entire dataset. As such, every iteration is very costly and not practical at all, given the training data set can be very large.

The paper considers Polyak-Lojasiewicz's condition, a more general form than strongly convex loss function. However, I find some of the comparison misleading. For example, Feldman et al. (STOC 2020) and Zhou et al. show an excess "population loss" of order $O(1/n)$. On the other hand, if I understand the paper correctly, the current submission only give excess empirical risk. Further, Feldman et al's two algorithms achieve optimal excess loss while running in nearly "linear" time, which is not the case in the current submission.

There might be some interesting idea in this paper, especially, looking at the dynamic schedule for PPML. However, I fail to see the strength of the result in the view of prior work as the comparison done in the paper is between apples and oranges.

I did not verify the proof very carefully. There are some typos that I found. For example, line following equation (1), $v_t$ should be $\nu_t$.

$\alpha$ depends on $f(\theta_t)- f(\theta^*)$ and $T, \sigma_t$ depends on $\alpha$. How are we going to set these parameters when we do not know the minimum value of the objective function?

---

> ### Author Response · Authors · 2020-11-16
> **Response to Reviewer #2**
>
>
> **Q1**: _My first objection to the paper is that gradient descent requires computation of gradients over the entire dataset. As such, every iteration is very costly and not practical at all, given the training data set can be very large._
>
> **A1**: We extend our results to private stochastic gradient descent (PSGD) in Sec. 4.3. It turns out that all the conclusions still hold for the stochastic gradients. Note the $T$ means the number of iterations rather than the epochs. Therefore, for large-scale problems, the PSGD provides a practical alternative to the PGD.
>
> **Q2**: _However, I find some of the comparison misleading. For example, Feldman et al. (STOC 2020) and Zhou et al. show an excess "population loss" of order O(1/n). On the other hand, if I understand the paper correctly, the current submission only gives excess empirical risk._
>
> **A2**: Thanks for pointing out this. We mark the difference in Table 1 of our revision to avoid misleading. But in this paper, we follow common practice in PGD convergence study (Bassily et al., 2014; Wang et al., 2017; Zhou et al., 2020). In the future, we are glad to study population loss as an extension of current work.
>
> **Q3**: _Further, Feldman et al's two algorithms achieve optimal excess loss while running in nearly "linear" time, which is not the case in the current submission._
>
> **A3**: After extending to the PSGD, we could greatly reduce the complexity of the algorithm. Moreover, our work focus on improving the utility within practical computation cost. Since our bound ($1/N^2$) is tighter than Feldman et al.'s ($1/N$ or $\log(N)/N$), we believe the PSGD with a dynamic schedule is still a powerful method to trade-off privacy and utility.
>
> **Q4**: _I fail to see the strength of the result in the view of prior work as the comparison done in the paper is between apples and oranges._
>
> **A4**: We compare our convergence bound to the line of recent works including uniform and dynamic methods. For example in Table 2, GD+Adv, GD+MA are the traditional uniform schedule PGD. And SSGD+zCDP uses dynamic batch size. Adam+MA uses a uniform schedule but with a momentum method where the moving average is kind of dynamic. To our best knowledge, these methods are representative in the sense of convergence analysis.
>
> As compared to prior works, The strength of using a dynamic schedule is on the factor of $N$. Our method has the tightest bound with $1/N^2$ factor while the others are either $1/N$ or $\log (N)/N$.
>
> **Q5**: _... line following equation (1), $v_t$ should be $\nu_t$._
>
> **A5**: Sorry for using such two similar symbols, but this is not a typo. $\nu_t$ represents the noise and $v_t$ is the updates to the model.
>
> **Q6**: _$\alpha$ depends on $f(\theta_t) − f(\theta^∗)$ and $T$, $\sigma_t$ depends on $\alpha$. How are we going to set these parameters when we do not know the minimum value of the objective function?_
>
> **A6**: The analysis does not directly give the solution of the optimal loss. Instead, we show how fast and to what extent the private gradient descent can converge. $f(\theta_t) − f(\theta^∗)$ is a constant independent from most learning algorithms. Thus, when comparing different algorithms, we only compare the non-constant part.

---

> > ### Comment · AnonReviewer2 · 2020-11-16
> > **On answers of the authors**
> >
> > On A1: I apologize, but I do not see any Section 4.3 in the original submission. Has it been changed post the reviews?
> > On A2: Bassily et al. also have a generalization bound. It is in the Appendix of their paper. Using uniform stability based argument, we get a better generalization bound (as used in Bassily et al. NeurIPS 2019).
> > On A3: This answer relies on the answer to A1.
> > On A4 and A5: Thanks for clarifying. I will have another pass over the submission.

---

> > > ### Author Response · Authors · 2020-11-17
> > > **About Sec 4.3**
> > >
> > > Thanks for your response.
> > > Yes. The Sec 4.3 is in the submission after the review. After reading your review, we agree SGD is important for the practice. Thus, we update our paper by adding a section for the Private SGD.

---

> > > > ### Author Response · Authors · 2020-11-20
> > > > **Updates on A2**
> > > >
> > > > Thanks for the references. We update the comparison in our latest revision (Nov. 19, 2020). Specifically, we used an additional table to compare the generalization error bounds, or true excess risk (TER) bounds, in Section A.1. In the new table, we also included Bassily et al.'s (2019) latest results on the generalization bounds. Using the same strategy from (Bassily et al., 2014), we extended our bounds to generalization bounds. Even though Bassily et al. achieved the generalization bounds close to non-private, we showed that our method still maintains the advantage when the full batch is used for training. For the generalization bounds, our result is close to $1/N$ rates as the Bassily et al.'s. Therefore, the results still showed the effectiveness of dynamic schedules on improving the convergence rates.

---

> > > > > ### Comment · AnonReviewer2 · 2020-11-20
> > > > > **Suggestion on A2**
> > > > >
> > > > > I believe that instead of going through the generalization trick used in Bassily et al. FOCS paper, a much better trick is to use the uniform stability based argument used in subsequent papers. If I remember correctly at the top of my head, it is in the last year NeurIPS paper by Bassily et al.
> > > > > I have not yet seen the updated version because I want to carefully see the SGD part as well (so you might see an updated comment). My current concern is that when we go to the generalization bound, we would get 1/N term that would be the higher order term and would take away the improvement in your result.

---

> > > > > > ### Author Response · Authors · 2020-11-22
> > > > > > **Updates on A2: Our bounds require fewer iterations both for GD and SGD**
> > > > > >
> > > > > > Greatly appreciate your prompt response and detailed comments. In our latest revision, we applied both tricks to extend our bounds to generalization bounds in Appendix A.1. When deriving the bounds, we note that the limitation of the true risk bounds lies on the extension from empirical bounds (either using uniform stability or the high-probability extension).  For uniform stability bounds, an important factor is the iteration number $T$, and we want to emphasize that our method has remarkably improved efficiency in $T$, from $N$ or $N^2$ to $\ln(N)$ (See Table 3). The improvement implies fewer iterations are required for converging to the same generalization error.
> > > > > >
> > > > > > Additionally, note our conclusion for private SGD is that the bounds do not get worse when using SGD because the effect of mini-batch is also $1/N^2$. Thus, the results in Table 3 can also be applied to SGD, which is more efficient in space complexity (compared to GD).

---

> > > > > > ### Author Response · Authors · 2020-11-23
> > > > > > **[continued] Updates on A2: improve bounds by uniform stability.**
> > > > > >
> > > > > > After reviewing the derivation of (Bassily et al. 2020) again, we found a way to further tighten our bound, by following its strategy to drop the term $G^2 \eta T /N \le O(G^2 \ln (N)/(MN) ) \le O (G^2/M)$. Therefore, we can obtain the true risk bounds directly from empirical bounds *without degradation*. Please check Appendix A.1 and Table 3 in our latest revision.
> > > > > >
> > > > > > In conclusion, our generalization bounds are as tight as the empirical bounds, with an efficient iterations of $\ln N$. Namely, we get $1/N^2$ utility bounds by dynamic schedules in $\ln N$ iterations.

---

### Official Review · AnonReviewer1 · 2020-10-29
**Review for "On Dynamic Noise Influence in Differential Private Learning"**

**Rating:** 5
**Confidence:** 4

**Review:**

Summary: The paper investigates the idea that a dynamic privacy schedule of decreasing noise can help private gradient descent. The main contribution is a theoretical analysis of a dynamic privacy schedule which helps reduce the utility upper bound of private gradient descent (with or without momentum).

Strengths:
1)	Private gradient descent is an important optimization technique in differentially private ML, therefore, understanding its behavior is broadly interesting.
2)	The paper investigates both the vanilla private gradient descent as well as a version with momentum.

Concerns:
1)	While there is some improvement with the dynamic schedule vs. fixed schedule, the improvement does not seem to significant (ref. Table 1).
2)	Results hold only under PL condition (which is a more general condition than strong convexity) and smoothness.

The results look correct. My current scores are because I see limited value/interest in these results.

Questions:
- Maybe relaxing the assumptions might be a way to strengthen the paper.  Is that possible?
- What is the advantage of using the momentum method in terms of the utility upper bound (ref. Table 1). It looks like (within constant factors) vanilla GD with dynamic schedule works as well as using the momentum.
- Do these results hold with a stochastic gradient?

Minor comment: hat{T} not defined in Table 1

---

> ### Author Response · Authors · 2020-11-16
> **Response to Reviewer #1**
>
> **Q1**: _While there is some improvement with the dynamic schedule vs. fixed schedule, the improvement does not seem to significant (ref. Table 1)._
>
> **A1**:
> 1. Our bound is $1/N^2$ and much tighter than other bounds, $1/N$, $\log (N)/N^2$ or $\log (N)/N$.
> 2. Our upper bound is closest to the non-private bound (within finite iterations) in terms of the sample efficiency.
> 3. Also, we are the first to provide a comprehensive analysis of the dynamic schedule which was only empirically studied by previous methods. We show to what extent the dynamic schedule can improve the utility. We believe the results will be a useful guide for developing adaptive or non-adaptive schedules for the private gradient descents. For example, if there is a moderate-size training set, the advantage of $\log (N)/N^2$ (dynamic schedule) versus $1/N^2$ (uniform schedule) can be significant. Then developing a dynamic schedule could be beneficial.
>
> **Q2**: _Results hold only under PL condition (which is a more general condition than strong convexity) and smoothness._
>
> **A2**:
> 1. The PL condition is a common assumption when studying the optimization problems (Karimi et al., 2016; Nesterov & Polyak, 2006; Reddi et al., 2016). For the PGD, it was already used to reveal the convergence properties (Wang et al. 2017; Zhou et al. 2020).
> 2. Moreover, the condition can be generalized to other cases, e.g., the strong convexity. And (Karimi et al., 2016) shows the following chain of implication: Strong Convex => Essential Strong Convexity => Weak Strongly Convexity => Restricted Secant Inequality => Polyak-Lojasiewicz Inequality. That means PL condition is general for many practical problems.
> 3. For non-convex problems, we empirically show the advantage of dynamic schedules as proved by our theorems.
>
> **Q3**: _Maybe relaxing the assumptions might be a way to strengthen the paper. Is that possible?_
>
> **A3**: We believe this is possible because we already empirically see that the dynamic schedule performs better than the uniform schedule in deep learning (Yu et al., 2019 S&P). In the future, we will continue to work on relaxing the assumptions.
>
> **Q4**: _What is the advantage of using the momentum method in terms of the utility upper bound (ref. Table 1). It looks like (within constant factors) vanilla GD with dynamic schedule works as well as using the momentum._
>
> **A4**: We show that the momentum method without a dynamic schedule results in a similar bound to the dynamic schedule for PGD, though the improvement disappears when $T$ is larger than some specific constant, $\hat T$. Therefore, the momentum method is a nice alternative to the PGD with a dynamic schedule. Moreover, the momentum method is more practical because its parameter $\beta$ is less dependent on the problem while the dynamic schedule highly depends on the loss curvature (ref. Lemma 4.1).
>
> **Q5**: _Do these results hold with a stochastic gradient?_
>
> **A5**: Yes. In the new version of the paper, we extend the gradient descent results to the stochastic gradient descent in Section 4.3. All the conclusions still hold for the stochastic gradients. Note the $T$ means the number of iterations rather than the epochs.

---

> > ### Comment · AnonReviewer1 · 2020-11-23
> > **On Authors Responses**
> >
> > Thanks for your responses. I think the inclusion of  the results stochastic gradient descent (Section 4.3) and the generalization results (Section A.1) definitely strengthen the paper.
> > Quick questions about the generalization results:
> > 1) In Table 3, are the results of SSGD+zCDP (Feldman et al. 2020) for general convex functions or strongly convex?
> > 2) The results of Wang et al. 2017 (which is essentially the static schedule) can also be plugged into the uniform stability bound of Bassily et al. 2020 to get the (almost) the same results. So why is this not in Table 3?

---

> > > ### Author Response · Authors · 2020-11-23
> > > **Reply to following questions**
> > >
> > > Thank your for your reply. Please find the reponses to your questions below.
> > >
> > > Q1: _In Table 3, are the results of SSGD+zCDP (Feldman et al. 2020) for general convex functions or strongly convex?_
> > >
> > > It is stong convexity, as mentioned in the caption of Table 3. For the general convex case, their bound will be mainly off by a $\ln N$, which does not change the conclusion from the comparison. For a fair comparison, we try to use the same assumption consistently in the table.
> > >
> > > Q2: _The results of Wang et al. 2017 (which is essentially the static schedule) can also be plugged into the uniform stability bound of Bassily et al. 2020 to get the (almost) the same results. So why is this not in Table 3?_
> > >
> > > Thanks for the advice. Please check our latest revision (Table 3) today where we just update the uniform stability bounds and (Wang et al. 2017). After reviewing the derivation of (Bassily et al. 2020) again, we found a way to further tighten our bound, by following its strategy to drop the term $G^2 \eta T /N \le O(G^2 \ln (N)/(MN) ) \le O (G^2/M)$. Therefore, we can obtain the true risk bounds directly from empirical bounds *without degradation*
> > >
> > > In conclusion, our generalization bounds are as tight as the empirical bounds, with an efficient iterations in $\ln N$. Namely, we get $1/N^2$ utility bounds by dynamic schedules in $\ln N$ iterations.

---

### Official Review · AnonReviewer3 · 2020-10-29
**This paper analyzes private gradient descent when using dynamic noise schedules.  They propose a dynamic schedule that minimizes the excess risk upper bound, and that the resulting bound on excess risk improves over the standard uniform schedule.**

**Rating:** 7
**Confidence:** 3

**Review:**

Strong Points:

1. The problem studied is interesting and important for the privacy community - private gradient descent is an important mechanism for private convex optimization, and one of the only mechanisms for non-convex optimization.
2. The authors do a good job engaging with prior work, identifying gaps in related work (i.e., theoretical justification for dynamic noise schedules) and putting their contributions in that context.
3. The analysis seems technically correct and pretty strong, although I did not check this closely.

Weak Points:
1. Excess risk upper bound is probably quite loose, so improving that may not necessarily improve actual excess risk.
2. log(N) improvement may be an artifact of the analysis, and not an inherent advantage of dynamic schedules.  Needs more clarification (see below).

Other Notes:

It’s not clear to me where the log(N) term in the excess risk upper bound comes from for the uniform noise schedule.  In other works on this topic, it is not there (e.g., https://arxiv.org/pdf/2005.04763.pdf) .  Since we are dealing with upper bounds, I wonder if it is just an artifact of the analysis of Wang et al., rather than an improvement offered by dynamic scheduling.  Are there other ways to remove this dependence other than dynamic noise?

---

> ### Author Response · Authors · 2020-11-16
> **Response to Reviewer #3**
>
> **Q1**: _Excess risk upper bound is probably quite loose, so improving that may not necessarily improve actual excess risk._
>
> **A1**: (Excess) risk upper bound is widely used to understand the utility of learning algorithms. Both the non-private algorithms, for example, momentum method under convex condition (Nesterov & Polyak, 2006) or private algorithms (Wang et al., 2017), (Feldmanetal.,2020) leverage the upper bound to study their advantages. Specifically, the upper bound could reveal the potential or the limitation of the private learning algorithms. Both previous and our work (Yu, et al., 2019 and Lee & Kifer, 2018) have empirically shown that a decreasing dynamic schedule could improve learning performance. Our work fills the theoretic gap to explain _why and to what extent_ this class of methods could improve Private Gradient Descent.
>
> **Q2**: _$log(N)$ improvement may be an artifact of the analysis, and not an inherent advantage of dynamic schedules. Needs more clarification (see below)._
>
> **A2**: To our best knowledge, it is highly likely that The $log(N)$ is not an artifact of Wang et al. since it also appears in other baselines (see Table 2 or 1). Regardless of the $\log (N)$ term, our bound has a $1/N^2$ factor which is smaller than most baselines which are either $1/N$ or $\log (N)/N$ except (Wang et al., 2017) (ref. Table 1). Moreover, our method is closest to the non-private bound if within the same iterations. Regarding the concern of the choice of the number of iteration, we emphasize that it is derived by minimizing the utility upper bound using the uniform schedule (please refer to the proof of Theorem 4.2).
>
> **Q3**: _It’s not clear to me where the $log(N)$ term in the excess risk upper bound comes from for the uniform noise schedule._
> _..._
> _Since we are dealing with upper bounds, I wonder if it is just an artifact of the analysis of Wang et al., rather than an improvement offered by dynamic scheduling. Are there other ways to remove this dependence other than dynamic noise?_
>
> **A3**: Notice for the uniform schedule, there is a T term in the utility upper bound, $\gamma^T + \alpha \kappa (1 - \gamma^T) T$ (See the sketch of the proof of Thm 4.2).  Because $T$ is proportional to the $\log{1\over \alpha} \propto \log (N)$ in Eq. (4), we see a $\log (N)$ in the final bound. The choice of $T$ is initially inspired by (Wang et al., 2017) where the reason is not given though. To support the usage of $T$, we minimize the utility upper bound w.r.t. $T$ and get a similar conclusion.
>
> When we use the dynamic schedule, the T term is replaced by the $\gamma^T$ at the beginning of the proof of Thm. 4.3. Because $\gamma<1$, $\gamma^T$ will be smaller than $T$ given the same $T$ value. Thanks to the exponential $\gamma^T$, there will be no $\log(N)$ any longer.
>
> We believe a dynamic schedule may not be the only way to solve the problem, just like many other optimization problems. For example, in our paper, we also provide an alternative way to reduce the upper bound: the momentum method. We show in Table 1 the momentum method with a uniform schedule has a similar upper bound to the PGD with a dynamic schedule. But the advantage by momentum only maintains for $T$ not larger than some constant $\hat T$. We are glad to further explore alternative ways to solve the problems in our future work.
>
> **Q4**: _In other works on this topic, it is not there (e.g., https://arxiv.org/pdf/2005.04763.pdf)._
>
> **A4**: The $\log (N)$ term also appears in prior works. (Feldman et al.), i.e., https://arxiv.org/pdf/2005.04763.pdf. Because the paper is formally published in STOC, thus here we only discuss the STOC version. For Feldman et al.’s work, the $\log N$ term appears in their strongly convex case. They do not have a $\log N$ for the convex case.  However, both bounds are not tighter than ours in terms of the $N$ because $\log (N)/N > 1/N > 1/N^2$. That means DPSGD with a dynamic schedule requires fewer samples to converge the same level of loss than the (Feldman et al.).

---

### Author Response · Authors · 2020-11-24
**Summary of responses**

We greatly appreciate the detailed and constructive comments from all reviewers.  With the help from reviewers' comments and suggestions, we have greatly improved our paper as summarized below:

* We extended the utility analysis from full-batch gradients to stochastic variants using mini-batch gradients, which are more practical for large-scale problems.
* Beyond the analysis on empirical risks, we also extended our work to analyze generalization errors or true risk bounds.

In our latest revision, we showed that our results (Table 1&2) are extensible for SGD (Section 4.3) and generalization error (Table 3) without degradation on corresponding bounds. In a nutshell, Private Gradient Descent (PGD) with dynamic schedules are shown to have practically and efficiently reduced the empirical loss on private datasets, as well as generalization loss on unseen data subject to the identical distribution. Specifically, our bounds have similar rates as the non-private ones and are tighter than state-of-the-art results only using iterations of order $\ln N$.

---

### Decision · Program_Chairs · 2021-01-07
**Final Decision**

**Decision:**

Reject

**Comment:**

This work proposes algorithms for solving ERM with continuous losses satisfying the PL condition. The first algorithm achieves that by using a chainging noise variance and thus the paper frames the contribution in terms of the advantages of non-constant noise rate.

The problem is a well-studied one and the result is a nice if relatively modest improvement over Wang et al. However, as pointed out in reviews, in the context of convex optimization the same rate has already been established (Feldman,Koren,Talwar STOC 2020). This work is cited and briefly discussed but the discussion only includes one of the algorithms in the paper (that does have an additional log N factor). The overall assumptions in this paper are not comparable (weaker in some ways and stronger since they only require PL instead of strong convexity) but still the overall the contribution appears to be incremental.